# Towards Reliable Simulation-Based Inference with Balanced Neural Ratio Estimation

**Arnaud Delaunoy**[*]
University of Liège
a.delaunoy@uliege.be

**Joeri Hermans**[*]
Unaffiliated
joeri@peinser.com

**François Rozet**
University of Liège
francois.rozet@uliege.be

**Antoine Wehenkel**
University of Liège
antoine.wehenkel@uliege.be

**Gilles Louppe**
University of Liège
g.louppe@uliege.be

## Abstract

Modern approaches for simulation-based inference rely upon deep learning surrogates to enable approximate inference with computer simulators. In practice, the estimated posteriors' computational faithfulness is, however, rarely guaranteed. For example, Hermans et al. [1] show that current simulation-based inference algorithms can produce posteriors that are overconfident, hence risking false inferences. In this work, we introduce Balanced Neural Ratio Estimation (BNRE), a variation of the NRE algorithm [2] designed to produce posterior approximations that tend to be more conservative, hence improving their reliability, while sharing the same Bayes optimal solution. We achieve this by enforcing a balancing condition that increases the quantified uncertainty in small simulation budget regimes while still converging to the exact posterior as the budget increases. We provide theoretical arguments showing that BNRE tends to produce posterior surrogates that are more conservative than NRE's. We evaluate BNRE on a wide variety of tasks and show that it produces conservative posterior surrogates on all tested benchmarks and simulation budgets. Finally, we emphasize that BNRE is straightforward to implement over NRE and does not introduce any computational overhead.

## 1 Introduction

Many areas of science and engineering use parametric computer simulations to describe complex stochastic generative processes. In this setting, Bayesian inference provides a principled framework to identify parameters matching empirical observations. Computer simulations, however, define the necessary likelihood function only implicitly, which prevents its evaluation and the use of classical inference algorithms. To overcome this obstacle, recent simulation-based inference (SBI) algorithms [3] build upon deep learning surrogates to approximate parts of the Bayes rule and enable approximate inference. For example, [4, 5] build a surrogate of the likelihood function while [6, 7, 2, 8, 9] approximate the likelihood-to-evidence ratio. The posterior can also be targeted directly with variational inference, as proposed by [10, 11, 5]. These algorithms are either amortized or run sequentially to drive the training towards a target observation and improve the simulation efficiency of the procedure [10, 12, 2, 11, 4, 8, 5]. However, sequential methods have the drawback of being computationally expensive to diagnose as the surrogates are only valid for the target observation [1]. Truncated marginal neural ratio estimation [9] alleviates this issue by introducing a sequential algorithm that builds a surrogate valid in a local region around the target.

---

[*]Equal contribution

36th Conference on Neural Information Processing Systems (NeurIPS 2022).

Since modern simulation-based inference algorithms rely on deep learning surrogates, concerns naturally arise regarding their computational faithfulness and whether they are sufficiently adequate for the inference task of interest. In Bayesian inference, these concerns can be at least partially addressed with diagnostics designed to probe the correct behaviour of the inference method, such as $\hat{R}$ diagnostics for MCMC [13], or to assess the quality of posterior approximations directly. The latter include diagnostics such as simulation-based calibration (SBC) [14] or coverage-based diagnostics [15, 1]. As discussed by Hermans et al. [1], posterior approximations must be conservative to guarantee reliable inferences, even when approximations are not faithful. For example, in the physical sciences, where the goal is often to constrain parameters of interest, wrongly excluding plausible values could drive the scientific inquiry in the wrong direction, whereas failing to exclude implausible values because of (too) conservative estimations is much less detrimental. Unfortunately, the same authors also demonstrate that current simulation-based inference algorithms can lead to overconfident surrogates and therefore false inferences.

Scientific use cases requiring conservative inference include for example the study of dark matter models in particle physics and astrophysics [16], which could be cold, warm, or hot dark matter. In general, thermal dark matter models are described by a single parameter, the dark matter thermal relic mass, which can be intuitively thought of as the energy the dark matter particle had in the Early Universe. Small values correspond to warm or hot dark matter, while high values are descriptive of cold dark matter. Applying an inference algorithm without diagnosing the learned estimator could lead to constraints that are tighter than they should be. For example, whenever an overconfident estimator produces posterior estimates that favor cold dark matter models, it could simultaneously reject alternative models, such as the extensively studied Sterile Neutrino, a potential candidate for the Warm Dark Matter particle. Making a scientific statement in this direction therefore requires the uttermost care to not wrongly exclude values of the thermal relic mass that are actually plausible.

In this work, we develop a novel algorithm that not only converges to exact inference as the simulation budget increases, but which is also more likely to produce conservative surrogates in small simulation budget regimes. Towards this objective, we propose a variant of the NRE algorithm called Balanced Neural Ratio Estimation (BNRE), which enforces a balancing condition on the binary neural classifier to increase the reliability of its posterior approximations.

The structure of the manuscript is outlined as follows. Section 2 describes the formalism and the necessary background. Section 3 describes BNRE and provides theoretical arguments towards its conservativeness and reliability. Section 4 illustrates our main results and provides insights regarding the behaviour of the method. Finally, Section 5 discusses related work while Section 6 summarizes our contributions and hints at future work.

## 2 Background

### 2.1 Statistical formalism

This work is concerned with simulation-based inference algorithms that produce posterior approximations $\hat{p}(\vartheta \,|\, x)$ under the following semantics. **Target parameters** $\vartheta$ denote the parameters of the model and we make the reasonable assumption that the prior $p(\vartheta)$ is tractable. The model is generically expressed as a computer program, a simulator, that describes the forward dynamics of interest based on the input parameters $\vartheta$. The simulator implicitly defines the likelihood function $p(x \,|\, \vartheta)$. While we cannot directly evaluate the likelihood $p(x \,|\, \vartheta)$, we can execute the computer program to generate synthetic **observables** $x \sim p(x \,|\, \vartheta)$. Every observable $x_o$ is tied to **ground truth** parameters $\vartheta^*$ whose forward evaluation within the simulator produced $x^*$.

Of special importance to Bayesians is the notion of a **credible region**, which is a domain $\Theta$ within the target parameter space that satisfies $\int_\Theta p(\vartheta \,|\, x = x^*)\, \mathrm{d}\vartheta = 1 - \alpha$ for some observable $x^*$ and confidence level $1 - \alpha$. Because many such regions exist, we target the credible region with the smallest volume, also known as the highest posterior density region [17, 18].

### 2.2 Neural ratio estimation

Neural Ratio Estimation (NRE) is an established approach in the simulation-based inference literature both from frequentist [6] and Bayesian [7, 2, 8, 9] perspectives. In essence, all protocols rely on

the density-ratio trick [19, 20, 6] to construct a surrogate of the likelihood ratio. In this work, we consider an amortized estimator $\hat{r}(\boldsymbol{x} \mid \boldsymbol{\vartheta})$ of the intractable likelihood-to-evidence ratio $r(\boldsymbol{x} \mid \boldsymbol{\vartheta}) = p(\boldsymbol{\vartheta}, \boldsymbol{x})/p(\boldsymbol{\vartheta})p(\boldsymbol{x}) = p(\boldsymbol{x} \mid \boldsymbol{\vartheta})/p(\boldsymbol{x})$ that can be learned by training a binary classifier $\hat{d} : \mathbf{X} \times \boldsymbol{\Theta} \mapsto [0, 1]$ to distinguish between samples of the joint $p(\boldsymbol{\vartheta}, \boldsymbol{x})$ with class label 1 and samples of the product of marginals $p(\boldsymbol{\vartheta})p(\boldsymbol{x})$ with class label 0, with equal label marginal probability. For the binary cross-entropy loss, the Bayes optimal classifier is

$$d(\boldsymbol{\vartheta}, \boldsymbol{x}) = \frac{p(\boldsymbol{\vartheta}, \boldsymbol{x})}{p(\boldsymbol{\vartheta}, \boldsymbol{x}) + p(\boldsymbol{\vartheta})p(\boldsymbol{x})} = \sigma \left( \log \frac{p(\boldsymbol{\vartheta}, \boldsymbol{x})}{p(\boldsymbol{\vartheta})p(\boldsymbol{x})} \right), \tag{1}$$

where $\sigma(\cdot)$ is the sigmoid function. Given target parameters $\boldsymbol{\vartheta}$ and an observable $\boldsymbol{x}$ supported by $p(\boldsymbol{\vartheta})$ and $p(\boldsymbol{x})$ respectively, the learned classifier $\hat{d}$ provides an approximation for the log likelihood-to-evidence ratio $\log r(\boldsymbol{x} \mid \boldsymbol{\vartheta})$ because $\log r(\boldsymbol{x} \mid \boldsymbol{\vartheta}) = \text{logit}(d(\boldsymbol{\vartheta}, \boldsymbol{x})) \approx \text{logit}(\hat{d}(\boldsymbol{\vartheta}, \boldsymbol{x})) = \log \hat{r}(\boldsymbol{x} \mid \boldsymbol{\vartheta})$. The log posterior density function is approximated as $\log \hat{p}(\boldsymbol{\vartheta} \mid \boldsymbol{x}) = \log p(\boldsymbol{\vartheta}) + \log \hat{r}(\boldsymbol{x} \mid \boldsymbol{\vartheta})$.

## 3  Balanced binary classification for neural ratio estimation

Following Hermans et al. [1], let us first define the **expected coverage probability** of the $1 - \alpha$ highest posterior density regions derived from the posterior estimator $\hat{p}(\boldsymbol{\vartheta} \mid \boldsymbol{x})$ as

$$\mathbb{E}_{p(\boldsymbol{\vartheta}, \boldsymbol{x})} \left[ \mathbb{1} \left( \boldsymbol{\vartheta} \in \Theta_{\hat{p}(\boldsymbol{\vartheta} \mid \boldsymbol{x})}(1 - \alpha) \right) \right], \tag{2}$$

where the function $\Theta_{\hat{p}(\boldsymbol{\vartheta} \mid \boldsymbol{x})}(1 - \alpha)$ yields the $1 - \alpha$ highest posterior density region of $\hat{p}(\boldsymbol{\vartheta} \mid \boldsymbol{x})$. This diagnostic probes the conservativeness of the posterior estimator (or the lack thereof) and can be interpreted as the expected frequentist coverage $\mathbb{E}_{p(\boldsymbol{\vartheta})}\mathbb{E}_{p(\boldsymbol{x} \mid \boldsymbol{\vartheta})} \left[ \mathbb{1} \left( \boldsymbol{\vartheta} \in \Theta_{\hat{p}(\boldsymbol{\vartheta} \mid \boldsymbol{x})}(1 - \alpha) \right) \right]$.

In this work, a posterior estimator has coverage at the confidence level $1 - \alpha$ whenever the expected coverage probability is larger or equal to the nominal coverage probability, $1 - \alpha$. We say that a posterior estimator is **conservative** when it has coverage for all confidence levels. The expected coverage probability can be plotted for various levels $\alpha$, which allows to visually identify conservative posterior estimators. The expected coverage can also be shown to be a special case of the SBC diagnostic [14] (see Appendix A), further motivating the usage of expected coverage.

Our main objective is to restrict the hypothesis space of the approximate classifiers $\hat{d}$ to those leading to conservative posterior estimators, hence solving the reliability concerns of NRE. Towards this goal, we construct a hypothesis space of **balanced classifiers** and show both theoretically and empirically that they lead to posterior estimators that tend to be more conservative.

### 3.1  Balanced binary classification

**Definition 1.** *A classifier $\hat{d}$ is balanced if* $\mathbb{E}_{p(\boldsymbol{\vartheta}, \boldsymbol{x})} \left[ \hat{d}(\boldsymbol{\vartheta}, \boldsymbol{x}) \right] = \mathbb{E}_{p(\boldsymbol{\vartheta})p(\boldsymbol{x})} \left[ 1 - \hat{d}(\boldsymbol{\vartheta}, \boldsymbol{x}) \right]$, *or*

$$\mathbb{E}_{p(\boldsymbol{\vartheta}, \boldsymbol{x})} \left[ \hat{d}(\boldsymbol{\vartheta}, \boldsymbol{x}) \right] + \mathbb{E}_{p(\boldsymbol{\vartheta})p(\boldsymbol{x})} \left[ \hat{d}(\boldsymbol{\vartheta}, \boldsymbol{x}) \right] = 1. \tag{3}$$

**Theorem 1.** *Any balanced classifier $\hat{d}$ satisfies* $\mathbb{E}_{p(\boldsymbol{\vartheta}, \boldsymbol{x})} \left[ \dfrac{d(\boldsymbol{\vartheta}, \boldsymbol{x})}{\hat{d}(\boldsymbol{\vartheta}, \boldsymbol{x})} \right] \geq 1.$

*Proof.* The integral form of the balancing condition

$$\iint \left( p(\boldsymbol{\vartheta}, \boldsymbol{x}) + p(\boldsymbol{\vartheta})p(\boldsymbol{x}) \right) \hat{d}(\boldsymbol{\vartheta}, \boldsymbol{x}) \, \mathrm{d}\boldsymbol{\vartheta} \, \mathrm{d}\boldsymbol{x} = 1 \tag{4}$$

implies that $\left( p(\boldsymbol{x}, \boldsymbol{\vartheta}) + p(\boldsymbol{\vartheta})p(\boldsymbol{x}) \right) \hat{d}(\boldsymbol{\vartheta}, \boldsymbol{x})$ is a valid density, both integrating to 1 and positive everywhere. Therefore, its Kullback-Leibler (KL) divergence with $p(\boldsymbol{\vartheta}, \boldsymbol{x})$ is positive. Through

Jensen's inequality, we obtain

$$0 \leq \mathrm{KL}\left(p(\boldsymbol{\vartheta}, \boldsymbol{x}) \,||\, (p(\boldsymbol{\vartheta}, \boldsymbol{x}) + p(\boldsymbol{\vartheta})p(\boldsymbol{x}))\hat{d}(\boldsymbol{\vartheta}, \boldsymbol{x})\right)$$

$$\leq \mathbb{E}_{p(\boldsymbol{\vartheta}, \boldsymbol{x})}\left[\log \frac{p(\boldsymbol{\vartheta}, \boldsymbol{x})}{(p(\boldsymbol{\vartheta}, \boldsymbol{x}) + p(\boldsymbol{\vartheta})p(\boldsymbol{x}))\hat{d}(\boldsymbol{\vartheta}, \boldsymbol{x})}\right]$$

$$\leq \mathbb{E}_{p(\boldsymbol{\vartheta}, \boldsymbol{x})}\left[\log \frac{d(\boldsymbol{\vartheta}, \boldsymbol{x})}{\hat{d}(\boldsymbol{\vartheta}, \boldsymbol{x})}\right]$$

$$\Rightarrow \quad 1 \leq \mathbb{E}_{p(\boldsymbol{\vartheta}, \boldsymbol{x})}\left[\exp\left(\log \frac{d(\boldsymbol{\vartheta}, \boldsymbol{x})}{\hat{d}(\boldsymbol{\vartheta}, \boldsymbol{x})}\right)\right] = \mathbb{E}_{p(\boldsymbol{\vartheta}, \boldsymbol{x})}\left[\frac{d(\boldsymbol{\vartheta}, \boldsymbol{x})}{\hat{d}(\boldsymbol{\vartheta}, \boldsymbol{x})}\right]. \qquad \square$$

**Theorem 2.** *Any balanced classifier $\hat{d}$ satisfies* $\mathbb{E}_{p(\boldsymbol{\vartheta})p(\boldsymbol{x})}\left[\dfrac{1 - d(\boldsymbol{\vartheta}, \boldsymbol{x})}{1 - \hat{d}(\boldsymbol{\vartheta}, \boldsymbol{x})}\right] \geq 1$.

*Proof.* Similar to Theorem 1, see Appendix B. $\qquad \square$

Theorem 1 shows that, in expectation over the joint distribution $p(\boldsymbol{\vartheta}, \boldsymbol{x})$, a balanced classifier $\hat{d}$ tends to make predictions whose probability values $\hat{d}(\boldsymbol{\vartheta}, \boldsymbol{x})$ are smaller than the exact probability values $d(\boldsymbol{\vartheta}, \boldsymbol{x})$. In other words, a balanced classifier $\hat{d}$ tends to be less confident than the Bayes optimal classifier $d$. Similarly, Theorem 2 shows that, in expectation over the product of the marginals $p(\boldsymbol{\vartheta})p(\boldsymbol{x})$, a balanced classifier tends to make predictions whose probability values $1 - \hat{d}(\boldsymbol{\vartheta}, \boldsymbol{x})$ are smaller than the exact probability values $1 - d(\boldsymbol{\vartheta}, \boldsymbol{x})$, hence showing that a balanced classifier $\hat{d}$ tends to also be less confident than the Bayes optimal classifier $d$. We note however that these two theorems hold only in expectation, which implies that neither $\hat{d}(\boldsymbol{\vartheta}, \boldsymbol{x}) \leq d(\boldsymbol{\vartheta}, \boldsymbol{x})$ for all $\boldsymbol{\vartheta}, \boldsymbol{x}$ nor $1 - \hat{d}(\boldsymbol{\vartheta}, \boldsymbol{x}) \leq 1 - d(\boldsymbol{\vartheta}, \boldsymbol{x})$ for all $\boldsymbol{\vartheta}, \boldsymbol{x}$ can generally be guaranteed.

**Theorem 3.** *The Bayes optimal classifier $d(\boldsymbol{\vartheta}, \boldsymbol{x})$ is balanced.*

*Proof.* Replacing the Bayes optimal classifier

$$d(\boldsymbol{\vartheta}, \boldsymbol{x}) \triangleq \frac{p(\boldsymbol{\vartheta}, \boldsymbol{x})}{p(\boldsymbol{\vartheta}, \boldsymbol{x}) + p(\boldsymbol{\vartheta})p(\boldsymbol{x})} \tag{5}$$

in the integral form of the balancing condition, we have

$$\iint (p(\boldsymbol{\vartheta}, \boldsymbol{x}) + p(\boldsymbol{\vartheta})p(\boldsymbol{x}))d(\boldsymbol{\vartheta}, \boldsymbol{x}) \,\mathrm{d}\boldsymbol{\vartheta} \,\mathrm{d}\boldsymbol{x}$$

$$= \iint \frac{(p(\boldsymbol{\vartheta}, \boldsymbol{x}) + p(\boldsymbol{\vartheta})p(\boldsymbol{x}))\, p(\boldsymbol{\vartheta}, \boldsymbol{x})}{p(\boldsymbol{\vartheta}, \boldsymbol{x}) + p(\boldsymbol{\vartheta})p(\boldsymbol{x})} \,\mathrm{d}\boldsymbol{\vartheta} \,\mathrm{d}\boldsymbol{x}$$

$$= \iint p(\boldsymbol{\vartheta}, \boldsymbol{x}) \,\mathrm{d}\boldsymbol{\vartheta} \,\mathrm{d}\boldsymbol{x} = 1. \qquad \square$$

Theorem 3 states that the Bayes optimal classifier is balanced. Therefore, **minimizing the cross-entropy loss while restricting the model hypothesis space to balanced classifiers results in the same Bayes optimal classifier of Eqn. 1.**

### 3.2 Balanced neural ratio estimation

We now extend the NRE algorithm to enforce the balancing condition. The previous results show that enforcing the condition should result in more conservative classifiers $\hat{d}$ and therefore to dispersed posterior approximations. Let us first note that Theorem 1 can be expressed as $\mathbb{E}_{p(\boldsymbol{x})}[\mathbb{E}_{p(\boldsymbol{\vartheta} \mid \boldsymbol{x})}[d(\boldsymbol{\vartheta}, \boldsymbol{x})/\hat{d}(\boldsymbol{\vartheta}, \boldsymbol{x}]] \geq 1$, which can (ideally) be achieved when the inner expectation is larger than 1 for all $\boldsymbol{x}$. In this case, the classifier $\hat{d}$ will be such that $\hat{d}(\boldsymbol{\vartheta}, \boldsymbol{x}) \leq d(\boldsymbol{\vartheta}, \boldsymbol{x})$ in

regions of high posterior density. Then,

$$\frac{\hat{d}(\boldsymbol{\vartheta}, \boldsymbol{x})}{1 - \hat{d}(\boldsymbol{\vartheta}, \boldsymbol{x})} \leq \frac{d(\boldsymbol{\vartheta}, \boldsymbol{x})}{1 - d(\boldsymbol{\vartheta}, \boldsymbol{x})}, \text{ which is equivalent to } \hat{r}(\boldsymbol{x} \mid \boldsymbol{\vartheta}) \leq r(\boldsymbol{x} \mid \boldsymbol{\vartheta}), \tag{6}$$

and $\hat{p}(\boldsymbol{\vartheta} \mid \boldsymbol{x}) \leq p(\boldsymbol{\vartheta} \mid \boldsymbol{x})$ since $\hat{p}(\boldsymbol{\vartheta} \mid \boldsymbol{x}) = p(\boldsymbol{\vartheta})\hat{r}(\boldsymbol{x} \mid \boldsymbol{\vartheta})$. Similarly, Theorem 2 implies $1 - d(\boldsymbol{\vartheta}, \boldsymbol{x}) \geq 1 - \hat{d}(\boldsymbol{\vartheta}, \boldsymbol{x})$ in regions of high prior density, which results in $p(\boldsymbol{\vartheta} \mid \boldsymbol{x}) \leq \hat{p}(\boldsymbol{\vartheta} \mid \boldsymbol{x})$. Between those two opposite effects, the constraint on $\hat{p}(\boldsymbol{\vartheta} \mid \boldsymbol{x})$ that will dominate depends on whether $p(\boldsymbol{\vartheta} \mid \boldsymbol{x}) > p(\boldsymbol{\vartheta})$ or $p(\boldsymbol{\vartheta} \mid \boldsymbol{x}) < p(\boldsymbol{\vartheta})$. If $p(\boldsymbol{\vartheta} \mid \boldsymbol{x}) > p(\boldsymbol{\vartheta})$, then $\hat{p}(\boldsymbol{\vartheta} \mid \boldsymbol{x}) \leq p(\boldsymbol{\vartheta} \mid \boldsymbol{x})$, whereas if $p(\boldsymbol{\vartheta} \mid \boldsymbol{x}) < p(\boldsymbol{\vartheta})$ then $p(\boldsymbol{\vartheta} \mid \boldsymbol{x}) \leq \hat{p}(\boldsymbol{\vartheta} \mid \boldsymbol{x})$. Overall, imposing the balancing condition will therefore result in approximate posteriors that lie between the prior and the exact posterior, without being more confident than they should.

Practically, the balancing condition can be targeted through a regularization penalty. For the binary cross-entropy $\mathcal{L}[\hat{d}] \triangleq -\mathbb{E}_{p(\boldsymbol{\vartheta}, \boldsymbol{x})}[\log \hat{d}(\boldsymbol{\vartheta}, \boldsymbol{x})] - \mathbb{E}_{p(\boldsymbol{\vartheta})p(\boldsymbol{x})}[\log(1 - \hat{d}(\boldsymbol{\vartheta}, \boldsymbol{x}))]$ and given that the balancing condition only depends on samples from $p(\boldsymbol{x})p(\boldsymbol{\vartheta})$ and $p(\boldsymbol{x}, \boldsymbol{\vartheta})$, the full loss functional including the balancing condition can be expressed as

$$\mathcal{L}_b\left[\hat{d}\right] \triangleq \mathcal{L}\left[\hat{d}\right] + \lambda \left(\mathbb{E}_{p(\boldsymbol{\vartheta})p(\boldsymbol{x})}\left[\hat{d}(\boldsymbol{\vartheta}, \boldsymbol{x})\right] + \mathbb{E}_{p(\boldsymbol{\vartheta}, \boldsymbol{x})}\left[\hat{d}(\boldsymbol{\vartheta}, \boldsymbol{x})\right] - 1\right)^2, \tag{7}$$

where $\lambda$ is a (scalar) hyper-parameter controlling the strength of the balancing condition's contribution. The training procedure is summarized in Algorithm 1. Since a classifier is balanced if the balancing condition cancels out, $\lambda$ could, in principle, be set arbitrarily large. However, as the balancing condition is estimated via Monte Carlo sampling, setting $\lambda$ to a large value could impair the classifier's learning ability. We found that $\lambda = 100$ works well across many problem domains with varying simulation budgets.

---

**Algorithm 1** Training algorithm for Balanced Neural Ratio Estimation (BNRE).

---

| *Inputs:* | Implicit generative model $p(\boldsymbol{x} \mid \boldsymbol{\vartheta})$ (simulator) and prior $p(\boldsymbol{\vartheta})$ |
|---|---|
| *Outputs:* | Approximate classifier $\hat{d}_\psi(\boldsymbol{\vartheta}, \boldsymbol{x})$ parameterized by $\psi$ |
| *hyper-parameters:* | Balancing condition strength $\lambda$ (default = 100) and batch-size $n$ |

**repeat**
  Sample data from the joint $\{\boldsymbol{\vartheta}_i, \ \boldsymbol{x}_i \sim p(\boldsymbol{\vartheta}, \boldsymbol{x}), \ y_i = 1\}_{i=1}^{n/2}$
  Sample data from the marginals $\{\boldsymbol{\vartheta}_i, \ \boldsymbol{x}_i \sim p(\boldsymbol{\vartheta})p(\boldsymbol{x}), \ y_i = 0\}_{i=n/2+1}^{n}$
  $\mathcal{L}[\hat{d}_\psi] = -\frac{1}{n} \sum_{i=1}^{n} y_i \log \hat{d}_\psi(\boldsymbol{\vartheta}_i, \boldsymbol{x}_i) + (1 - y_i) \log(1 - \hat{d}_\psi(\boldsymbol{\vartheta}_i, \boldsymbol{x}_i))$
  $\mathcal{B}[\hat{d}_\psi] = \frac{2}{n} \sum_{i=1}^{n/2} \hat{d}_\psi(\boldsymbol{\vartheta}_i, \boldsymbol{x}_i) + \frac{2}{n} \sum_{i=n/2+1}^{n} \hat{d}_\psi(\boldsymbol{\vartheta}_i, \boldsymbol{x}_i)$
  $\psi = \texttt{minimizer\_step(params=}\psi\texttt{, loss=}\mathcal{L}[\hat{d}_\psi] + \lambda(\mathcal{B}[\hat{d}_\psi] - 1)^2)$
**until convergence**
**return** $\hat{d}_\psi(\boldsymbol{\vartheta}, \boldsymbol{x})$.

---

# 4 Experiments

We start by providing an extensive validation of BNRE on a broad range of benchmarks demonstrating that the proposed method alleviates the problem. Section 4.2 follows up with an illustrative demonstration on the behaviour of BNRE and its hyper-parameters. Code is available at https://github.com/montefiore-ai/balanced-nre.

## 4.1 Extensive validation

**Setup** We evaluate the expected coverage of posterior estimators produced by both NRE and BNRE on various problems. Those benchmarks cover a diverse set of problems from particle physics (Weinberg), epidemiology (Spatial SIR), queueing theory (M/G/1), population dynamics (Lotka Volterra, and astronomy (Gravitational Waves). They are representative of real scientific applications of simulation-based inference. A more detailed description of the benchmarks can be found in Appendix C. The architectures and hyper-parameters used for each problem are defined in Appendix

D. Our evaluation considers simulation budgets of increasing size, ranging from $2^{10} = 1024$ to $2^{17} = 131,072$ samples, and credibility levels from $0.05$ to $0.95$. For every simulation budget, we train 5 posterior estimators for 500 epochs and determine the credible region by evaluating the approximated posterior density function in a discretized and empirically normalized grid of the parameter space with sufficient resolution. The subsequent credible region is the set of parameters whose estimated (and normalized) posterior density is higher or equal to an inclusion threshold fitted to obtain the desired credibility level $1 - \alpha$. Details on this procedure are described in Appendix E. The expected coverage probability is estimated on 10000 unseen samples from the joint $p(\vartheta, x)$, for each considered credibility level.

**Expected coverage**   The expected coverage curves and their interpretation are detailed in Figure 1. We observe that NRE often produces posterior estimators that are overconfident, especially for small simulation budgets. However, NRE's reliability increases with the availability of training data. By contrast, **BNRE produces posterior estimators that are conservative on all benchmarks for all simulation budgets**. Figure 2 explores the same phenomena through a quantity which we call the coverage AUC, highlighting the effect of the simulation budget. Coverage AUC corresponds to the integrated signed area between the expected coverage curve and the diagonal of a particular simulation. From this quantity it is evident there is a clear distinction between NRE and BNRE with respect to the available simulation budget. Both methods have the tendency to converge towards 0, indicating both methods are moving closer to the Bayes optimal classifier. However, the difference between these methods lies with how this solution is approached. While NRE can approach this limit from both sides, BNRE consistently produces coverage AUC's above 0, corresponding to conservative posterior approximations, and therefore exhibits the desired behaviour (in expectation).

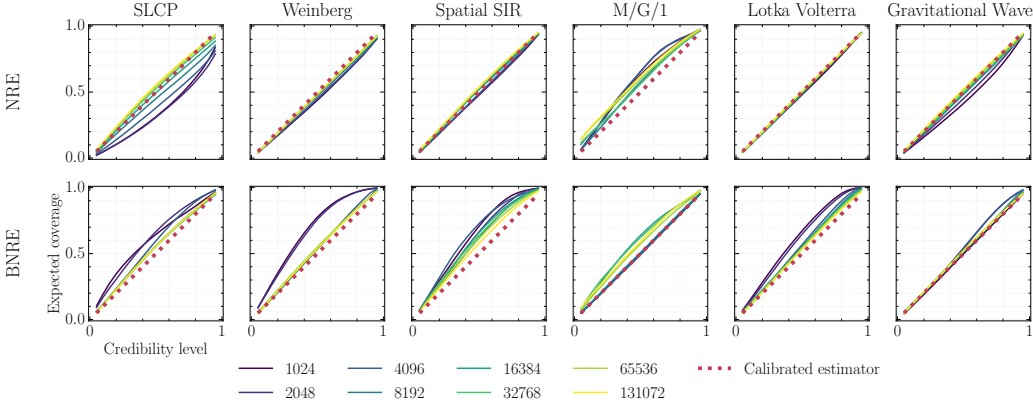

Figure 1: Expected coverage for increasing simulation budgets. A perfectly calibrated posterior has an expected coverage probability equal to the nominal coverage probability and hence produces a diagonal line. A conservative estimator has an expected coverage curve at or above the diagonal line, while an overconfident estimator produces curves below the diagonal line. The diagnostic therefore provides an immediate visual interpretation. We observe that NRE can produce overconfident estimators, while BNRE always produces coverage curves above the diagonal line and therefore the desired behaviour: conservative posterior approximations. The means over 5 runs are reported.

**Statistical performance**   In addition to the reliability of the posteriors, we evaluate and compare the statistical performance of the posterior approximations produced by NRE and BNRE. We estimate the expected approximate log posterior density $\mathbb{E}_{p(\vartheta, x)} \left[ \log \hat{p}(\vartheta \mid x) \right]$ over a large number of pairs $\vartheta, x$. It captures how well the posterior surrogates $\hat{p}(\vartheta \mid x)$ approximate the true posteriors $p(\vartheta \mid x)$ since $\mathbb{E}_{p(\vartheta, x)} \left[ \log \hat{p}(\vartheta \mid x) \right] = -\mathbb{E}_{p(x)} \text{KL} \left[ p(\vartheta \mid x) \mid\mid \hat{p}(\vartheta \mid x) \right] + \mathbb{E}_{p(x)} \mathbb{E}_{p(\vartheta \mid x)} \left[ \log p(\vartheta \mid x) \right]$ [21].

Figure 3 shows our results. We observe that enforcing the balancing condition for $\lambda = 100$ is associated with a loss in statistical performance. However, the loss in statistical performance is eventually recovered by increasing the simulation budget. In fact, practitioners might be inclined to favor reliability over statistical performance [1], although it is always a trade-off that depends on the use case. Nevertheless, it is possible to improve the statistical performance by tuning the surrogate, or by increasing the available simulation budget as we have demonstrated.

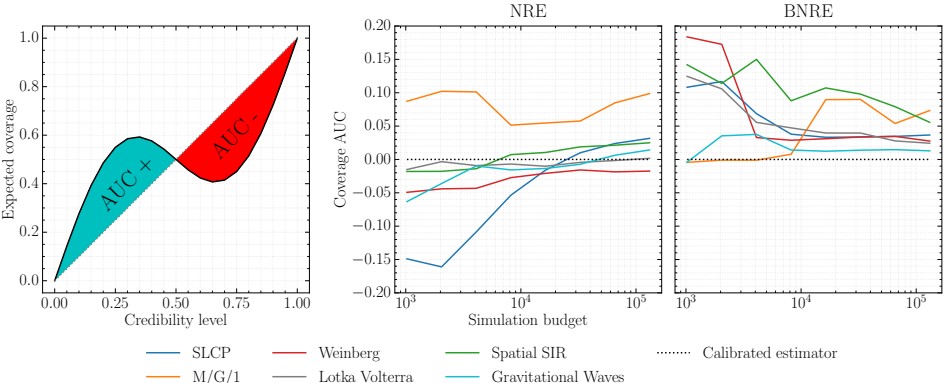

Figure 2: Coverage AUC measures the integrated signed area between the expected coverage curve and the diagonal. A perfectly calibrated posterior has an expected coverage probability equal to the nominal coverage probability, producing a diagonal line and has a coverage AUC of $0$, as shown on the left subplot. A conservative estimator on the other hand has a coverage AUC larger than $0$ and an overconfident estimator smaller than $0$. We observe that while NRE can produce coverage AUC both below or above $0$, BNRE always produces a coverage AUC larger than $0$, implying that its posterior approximations are conservative on average. The means over $5$ runs are reported. A complete overview, including standard deviations, are provided in Appendix F.

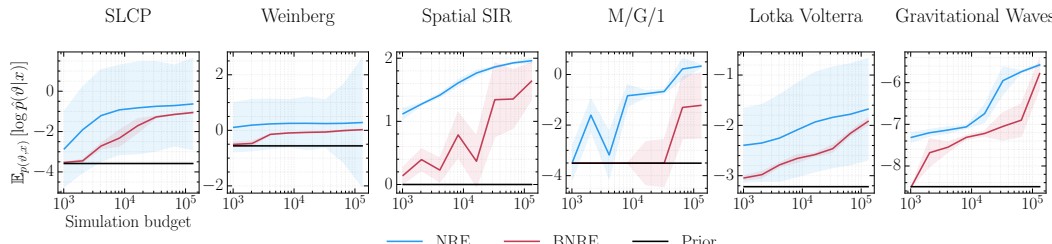

Figure 3: Expected value $\mathbb{E}_{p(\boldsymbol{\vartheta},\boldsymbol{x})}\big[\log \hat{p}(\boldsymbol{\vartheta} \,|\, \boldsymbol{x})\big]$ of the approximate log posterior density of the nominal parameters with respect to the simulation budget. We observe that BNRE produces log posterior densities lower than NRE. This shows that enforcing the balancing condition to have more reliable posterior approximates comes at the price of a small loss in information gain. However, BNRE improves over the prior and eventually converges towards NRE as the simulation budget increases. Solid lines represent the mean over $5$ runs and shaded areas represent the standard deviation.

## 4.2 In-depth analysis

In this section, we consider the Weinberg benchmark as described in Appendix C. The quality of the posterior approximations produced by BNRE is initially discussed with respect to the simulation budget. Afterwards, the effects of the hyper-parameter $\lambda$ are studied.

**Quality assessment** Because the expected coverage does not capture the quality of an approximation in terms of information gain, we complement our assessment with a bias and variance analysis of the posterior approximations. Let us consider the expected squared error over the approximate posterior $\mathbb{E}_{\hat{p}(\boldsymbol{\vartheta} \,|\, \boldsymbol{x})}\big[(\boldsymbol{\vartheta} - \boldsymbol{\vartheta}^*)^2\big]$, where $\boldsymbol{\vartheta}^*$ is the ground truth parameter value. With $\bar{\boldsymbol{\vartheta}}(\boldsymbol{x}) = \mathbb{E}_{\hat{p}(\boldsymbol{\vartheta} \,|\, \boldsymbol{x})}[\boldsymbol{\vartheta}]$, we decompose $\mathbb{E}_{\hat{p}(\boldsymbol{\vartheta} \,|\, \boldsymbol{x})}\big[(\boldsymbol{\vartheta} - \boldsymbol{\vartheta}^*)^2\big]$ as

$$\mathbb{E}_{\hat{p}(\boldsymbol{\vartheta} \,|\, \boldsymbol{x})}\Big[\big(\boldsymbol{\vartheta} - \bar{\boldsymbol{\vartheta}}(\boldsymbol{x})\big)^2\Big] + 2\big(\bar{\boldsymbol{\vartheta}}(\boldsymbol{x}) - \boldsymbol{\vartheta}^*\big) \underbrace{\mathbb{E}_{\hat{p}(\boldsymbol{\vartheta} \,|\, \boldsymbol{x})}\big[\big(\boldsymbol{\vartheta} - \bar{\boldsymbol{\vartheta}}(\boldsymbol{x})\big)\big]}_{=0} + \mathbb{E}_{\hat{p}(\boldsymbol{\vartheta} \,|\, \boldsymbol{x})}\Big[\big(\bar{\boldsymbol{\vartheta}}(\boldsymbol{x}) - \boldsymbol{\vartheta}^*\big)^2\Big]$$

$$= \mathbb{E}_{\hat{p}(\boldsymbol{\vartheta} \,|\, \boldsymbol{x})}\Big[\big(\boldsymbol{\vartheta} - \bar{\boldsymbol{\vartheta}}(\boldsymbol{x})\big)^2\Big] + \big(\bar{\boldsymbol{\vartheta}}(\boldsymbol{x}) - \boldsymbol{\vartheta}^*\big)^2.$$

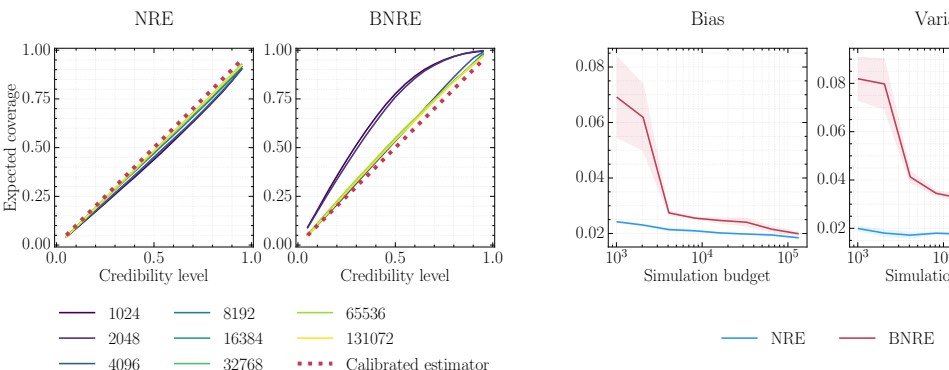

Figure 4: Comparison between NRE and BNRE in terms of expected coverage, bias and variance on the Weinberg benchmark. On the left side, the coverage is shown with respect to the simulation budget represented by the colormap. The bias and variance are represented on the right side of the plot. BNRE is run with $\lambda = 100$. Consistent with our previous observations in Figure 3, we observe that the gap in both bias and variance reduces as the simulation budget increases. Futhermore, in contrast with NRE, the posterior approximations of BNRE are tending towards being increasingly calibrated while at the same time being conservative. Solid lines represent the mean over 5 runs and shaded areas represent the standard deviation.

The expectation over the joint distribution $p(\boldsymbol{\vartheta}^*, \boldsymbol{x})$ of the expected squared error can hence be decomposed in a bias term defined as

$$\text{bias}(\hat{p}(\boldsymbol{\vartheta} \,|\, \boldsymbol{x})) \triangleq \mathbb{E}_{p(\boldsymbol{\vartheta}^*, \boldsymbol{x})} \left[ \left( \bar{\boldsymbol{\vartheta}}(\boldsymbol{x}) - \boldsymbol{\vartheta}^* \right)^2 \right], \tag{8}$$

which can be interpreted as the expected discrepancy between the nominal value $\boldsymbol{\vartheta}^*$ and the expected posterior value $\bar{\boldsymbol{\vartheta}}$. The variance term is

$$\text{variance}(\hat{p}(\boldsymbol{\vartheta} \,|\, \boldsymbol{x})) \triangleq \mathbb{E}_{p(\boldsymbol{\vartheta}^*, \boldsymbol{x})} \left[ \mathbb{E}_{\hat{p}(\boldsymbol{\vartheta} \,|\, \boldsymbol{x})} \left[ \left( \boldsymbol{\vartheta} - \bar{\boldsymbol{\vartheta}}(\boldsymbol{x}) \right)^2 \right] \right] \tag{9}$$

and measures the dispersion of the posterior approximations. Note that these terms differ from the typical statistical bias and variance of point estimators since we are considering full posterior estimators. In particular, the bias of the Bayes optimal model does not necessarily reduce to $0$.

Figure 4 shows the evolution of expected coverage, bias and variance with respect to the available simulation budget. By taking all plots into consideration with respect to the simulation budget, we can validate that – as suggested by theorems 1 and 2 – the increase in expected coverage is tied to an increase in variance. However, this increase comes at the price of a slight increase in bias. Consistent with our previous observations in Figure 3, we observe that the gap in both bias and variance reduces as the simulation budget increases. The bias gets close to $0$ for high simulation budgets, showing that the bias induced by BNRE vanishes as the simulation-budget increases. A bias and variance analysis for all remaining benchmarks is discussed in Appendix G.

**Effects of $\lambda$**   Finally, Figure 5 shows the effect the hyper-parameter $\lambda$ on the posterior approximations, their expected coverage and the balancing condition. BNRE is run 5 times for $\lambda$ ranging from 1 to $2^{15}$ and for a fixed simulation budget of $1024$. Initially, the effect on the posterior approximations is limited for small values of $\lambda$. However, once $\lambda$ increases, the balancing condition forces the posterior approximations to become increasingly dispersed and conservative. Eventually, at least for this specific simulation budget, the posterior approximation reduces to the prior as the balancing condition becomes dominant over the cross-entropy term. Although the global optimum remains unchanged as stated by Theorem 3, large $\lambda$ values are likely to impair the training procedure. In particular, a large $\lambda$ can inflate the statistical noise of the Monte Carlo estimation of the balancing condition and make the classifier $\hat{d}$ degenerate to a classifier that is trivially balanced such as the random classifier $\hat{d}(\boldsymbol{\vartheta}, \boldsymbol{x}) = 0.5$ for all $\boldsymbol{\vartheta}, \boldsymbol{x}$. In this case, $\hat{r}(\boldsymbol{x} \,|\, \boldsymbol{\vartheta}) = 1$ for all $\boldsymbol{\vartheta}, \boldsymbol{x}$ and the approximate posterior degenerates to the prior. This effect is directly evident from Figure 5, starting from $\lambda \simeq 1000$. In practice, $\lambda$ should be sufficiently large such that the approximate classifier is balanced,

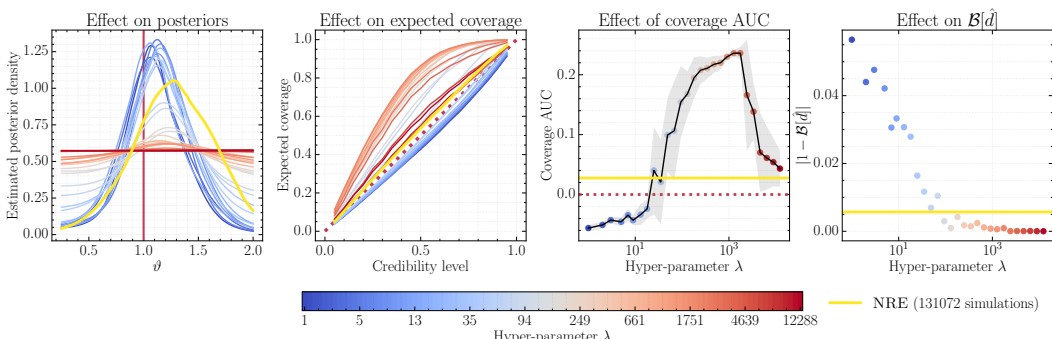

Figure 5: Effect of the hyper-parameter $\lambda$ for a fixed simulation budget of 1024. The first plot from left to right shows the evolution of the approximate posterior for a given observation at a fixed $\vartheta^*$, indicated by the red vertical line. This approximate posterior is compared to NRE trained on a large simulation budget, shown in yellow and serving as a proxy for the true posterior. The second plot illustrates the empirical expected coverage. The third plot provides a summarized view of the second plot using the coverage AUC as summary statistic. The fourth plot shows that classifiers are becoming increasingly more balanced as $\lambda$ increases. In addition, the plots show that $\lambda$ is directly tied to the statistical performance and reliability of the posterior approximations. Classifiers trained with small $\lambda$'s are associated with (relatively) tight posteriors and overconfident approximations, while classifiers trained with larger values of $\lambda$ are increasingly more dispersed and conservative until the posterior approximations reduce to the prior due to inflated statistical noise of the Monte Carlo estimation of the balancing condition. Furthermore, the expected coverage plot shows the estimator is almost perfectly calibrated and implicitly balanced. Immediately visible from the various posterior approximations in the leftmost subplot, is the fact that BNRE produces overconfident and biased approximations in the presence of a small simulation budget and a small $\lambda$, indicated by their dark blue color. However, the balancing condition can be applied to the underlying estimator to improve its reliability by increasing $\lambda$. Ideally, $\lambda$ should be as small as possible to maximize predictive performance, while at the same time remain sufficiently large to guarantee coverage. From the 3rd subplot from the left, in this particular problem setting, that happens at the point where the coverage AUC transitions from being negative to positive ($\lambda \approx 25.0$).

while maximizing the statistical performance of the posterior estimator. Therefore, we recommend to start with a small value for $\lambda$ and to gradually increase $\lambda$ until the posterior estimator becomes conservative. We empirically found $\lambda = 100$ to be a reasonably good default value leading to good performance across all considered benchmarks with various model architectures.

## 5 Related work

In the Bayesian setting, BNRE improves the reliability of NRE by constraining the classifier hypothesis space to balanced classifiers, which results in more conservative posteriors. Towards the same objective of conservative and reliable approximate posteriors, Hermans et al. [1] have shown empirically that ensembling posterior estimators increases their expected coverage. Since the two solutions are complementary, we suggest that ensembling BNRE is a safe practice to follow. To the best of our knowledge, no other related work exists to make Bayesian simulation-based inference algorithms more conservative and reliable.

In the frequentist setting, Cranmer et al. [6] make use of neural ratio estimation to learn likelihood ratio test statistics. They show that the classifier $\hat{d}$ does not need to be exact for the statistic to remain the most powerful, provided that the approximate likelihood ratio is monotonic with exact likelihood ratio. When this is not the case, robust inference remains possible by calibrating the classifier, at the price of a loss in statistical power. Similarly, for frequentist likelihood-free inference, Dalmasso et al. [22] use classifiers to estimate likelihood ratio statistics and propose a procedure for guaranteeing valid hypothesis tests and confidence sets. Finally, Dalmasso et al. [23] propose a practical procedure for the Neyman construction of confidence sets with finite-sample guarantees of nominal coverage as well as diagnostics that estimate conditional coverage over the entire parameter space.

In this work, we make the assumption that the simulator is well-specified, in the sense that it accurately models the real data generation process. However, this assumption is often violated. To overcome this issue, Generalized Bayesian inference (GBI) extends Bayesian inference by replacing the likelihood term by with arbitrary loss function [24]. Those loss functions can be designed to mitigate specific types of misspecifications and enable robust inference, even with intractable likelihoods [25–27]. Power likelihood losses have also been shown to increase robustness to model misspecification [28]. It consists in raising the likelihood to a power to control the impact it has over the prior. The lower the power of likelihood, the lower the importance given to the data and the higher the uncertainty of the posterior. It can either be set based on practitioner knowledge or derived from observed data [29]. Following the same objective, Miller and Dunson [30] introduce coarsened posteriors that condition on a neighborhood of the empirical data distribution rather than on the data itself. This neighborhood is derived from a distance function that, when set to the relative entropy, allows the approximation of coarsened posteriors by a power posterior. Recently, Dellaporta et al. [31] applied Bayesian non-parametric learning to SBI, making inference with misspecified simulator models both robust and computationally efficient.

## 6    Conclusions and future work

In this work, we introduced Balanced Neural Ratio Estimation (BNRE), a variation of neural ratio estimation designed to produce more conservative posterior estimators, even when the likelihood-to-evidence ratio estimator is not computationally faithful. We provide theoretical arguments suggesting that enforcing the balancing condition should lead to more conservative posteriors without sacrificing exactness in the large simulation budget regime. Our theoretical results are experimentally validated on benchmarks of varying complexity.

Nevertheless, our inference algorithm comes with limitations that practitioners should keep in mind. First, we emphasize that theorems 1 and 2 hold only in expectation, which means that we cannot provide any guarantee at the level of single inferences. Second, the balancing condition is enforced through a regularization penalty that is not estimated exactly. This implies that the classifier $\hat{d}$ is rarely strictly balanced, although close to be, in which case theorems 1 and 2 do not hold. Third, the benefits of BNRE remain to be assessed in high-dimensional parameter spaces. In particular, the posterior density must be evaluated on a discretized grid over the parameter space to compute credibility regions, which currently prohibits the accurate computation of expected coverage in the high-dimensional setting. In conclusion, BNRE **should not be viewed as a way to obtain conservative posterior estimators with 100% reliability, but rather as a way to increase the reliability of the posterior estimators with minimal effort and no computational overhead**.

Looking forward, the balancing condition could potentially be applied to other simulation-based inference algorithms. Future works could include a generalization to neural posterior estimation (NPE). In fact, the likelihood-to-evidence ratio can be extracted from an approximate posterior by removing its dependence on the prior, $\log \hat{r}(\boldsymbol{x} \,|\, \boldsymbol{\vartheta}) = \log \hat{p}(\boldsymbol{\vartheta} \,|\, \boldsymbol{x}) - \log p(\boldsymbol{\vartheta})$, which in turn can be expressed as a classifier $\hat{d}(\boldsymbol{\vartheta}, \boldsymbol{x}) = \sigma(\log \hat{r}(\boldsymbol{x} \,|\, \boldsymbol{\vartheta}))$ on which the balancing condition can be evaluated and enforced. Although our work focuses on amortized approximate inference, the balancing condition could also be applied to sequential inference algorithms to increase their reliability.

Finally, although our initial motivation is framed within the field of simulation-based inference, our theoretical results are directly applicable to **any binary classification task** by replacing the joint and marginal distributions in the balancing condition with the distributions of the two considered classes. Therefore, it provides an easy-to-implement modification for high-risk classification problems.

## Acknowledgments and Disclosure of Funding

Arnaud Delaunoy, Joeri Hermans and Antoine Wehenkel would like to thank the National Fund for Scientific Research (F.R.S.-FNRS) for their scholarships. Gilles Louppe is recipient of the ULiège - NRB Chair on Big Data and is thankful for the support of NRB. Computational resources have been provided by the Consortium des Équipements de Calcul Intensif (CÉCI), funded by the National Fund for Scientific Research (F.R.S.-FNRS) under Grant No. 2.5020.11 and by the Walloon Region.

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
