# OpenReview forum: "Towards Reliable Simulation-Based Inference with Balanced Neural Ratio Estimation"
_NeurIPS.cc/2022/Conference — NeurIPS 2022 Accept_

### Official Review · Reviewer_X2bu · 2022-06-26

**Rating:** 6
**Confidence:** 5
**Soundness:** 3 good
**Presentation:** 2 fair
**Contribution:** 3 good

**Summary:**

The paper introduces and discusses a regularization approach for neural ratio estimation that is based on enforcing a balancing condition of the classifier (via an additional regularisation term). It is shown theoretically that on average (expectation) this results in a more conservative classifier.
In several experiments it is shown that the approach indeed results in a more conservative posterior.

**Questions:**

- Could you please write down the definition of a classifier? I assume you mean something like $d: X \rightarrow [0, 1]$
- The discussion around equation (6) is confusing. You say that $\hat{p}(\theta|x) < {p}(\theta|x)$ whenever the $\hat{d}(\theta,x) < {d}(\theta,x)$, but clearly $\hat{p}(\theta|x) < {p}(\theta|x)$ can't hold for all $\theta$ since we are considering densities. Hence, the crucial aspects of whether the density is more conservative is _for which_ $\theta$ one has $\hat{p}(\theta|x) < {p}(\theta|x)$ and that seems to be missing.
- Generally the model details are a bit short. For example, you reference Lotka and Volterra indicating that the underlying model is a deterministic ODE following the Lotka-Volterra equations. However, commonly people in SBI work with the stochastic version, describing a Markov jump process. I would be nice to have more information here.
- I am not so sure about the expected coverage as a special case of SBC. The internal consistency of the joint distribution that you're relying on is something that is remarked even in the SBC paper to be known before. The devil is in the detail and from my point of view SBC is about an actionable algorithm, which is more than just the expectation that you allude to in Appendix A. In particular, the actual computations (checking whether rank statistics are uniformly distributed vs comparing coverage) are quite different in both cases.

**Limitations:**

The paper contains a good discussion of limitations. A slight improvement would be to discuss the trade-off between accurate calibration and conservative posteriors as mentioned above.

**Strengths And Weaknesses:**

__Strengths__
- Ensuring that posteriors derived by deep learning algorithms are valid is very important.
- The use of the balancing condition as an additional measure is an interesting idea.
- The method is easy to implement.
- The experiments show that the posterior tends to be more conservative when the regularizer is applied.
- The induced bias and variance diminish with increasing sample size
- Little to no exaggeration. The authors are very honest about the advantages and disadvantages of their method which makes the contribution all the more valuable to the field.

__Weaknesses__
- The penalty term does result in a worse calibration in most examples. While the posterior is more conservative the calibration is often worse, in particular when the number of samples is low. Especially for models where simulation is costly this might be an issue.
- While I agree that conservative posteriors are preferable, calibration is not a binary issue. Overly conservative posteriors might be unable to identify a parameter to a reasonable degree. In the Weinberg, SIR and Lotka Volterra models, a relatively accurate posterior is traded in for a mis-calibrated conservative posterior. I think the trade-off is important.
- There is a broad literature in statistics considering model (over-)confidence [A - C] which I think should be mentioned. In particular, power posteriors are quite popular to adjust the posterior in order to account for discrepancies between a model and real data. Since NRE does not estimate the likelihood directly, it might be necessary to use the presented approach instead of power posteriors, but a discussion should be added.

> Recently, Dellaporta et al. [26] further improved GBI by combining
- The cited work has advantages and disadvantaged but it's not an "improved" version of GBI, it's a variant. In particular, it best applies to IID data. When using time-series data as many of the models in this paper [D] would be a more fitting GBI procedure.

> M/G/1, originally introduced by Papamakarios et al. [4],
- Papamakarios et al. however reference [E] for the M/G/1 model. This should be amended.


__References__

[A] - Peter Grünwald. Safe learning: bridging the gap between Bayes, MDL and statistical learning
theory via empirical convexity. In Proceedings of the 24th Annual Conference on Learning
Theory, pages 397–420, 2011.

[B] - Peter Grünwald. The safe Bayesian. In International Conference on Algorithmic Learning
Theory, pages 169–183. Springer, 2012.

[C] - Chris Holmes and Stephen Walker. Assigning a value to a power likelihood in a general
bayesian model. Biometrika, 104(2):497–503, 2017

[D] - Dyer, Joel, Patrick Cannon, and Sebastian M. Schmon. "Approximate Bayesian Computation with Path Signatures." arXiv preprint arXiv:2106.12555 (2021).

[E] -  A. Y. Shestopaloff and R. M. Neal. On Bayesian inference for the M/G/1 queue with efficient MCMC
sampling. arXiv:1401.5548, 2014.

---

> ### Author Response · Authors · 2022-08-01
> **Rebuttal**
>
> Thank you for your positive review highlighting the importance, novelty, and significance of BNRE. We also appreciate your kind words regarding the presentation of our work.
>
> > The penalty term does result in a worse calibration in most examples. While the posterior is more conservative the calibration is often worse, in particular when the number of samples is low. Especially for models where simulation is costly this might be an issue.
>
> > While I agree that conservative posteriors are preferable, calibration is not a binary issue. Overly conservative posteriors might be unable to identify a parameter to a reasonable degree. In the Weinberg, SIR and Lotka Volterra models, a relatively accurate posterior is traded in for a mis-calibrated conservative posterior. I think the trade-off is important.
>
> We agree, the trade-off is important and can actually be adjusted by tuning the $\lambda$ parameter depending on the use case.
>
> We now better highlight and nuance the importance of the trade-off in our discussion of Figure 3. We have replaced the sentence `However, the loss in statistical performance is eventually recovered by increasing the simulation budget. In fact, practitioners might be inclined to favor reliability over statistical performance [1] and would therefore be willing to cover this cost.` by `However, the loss in statistical performance is eventually recovered by increasing the simulation budget. In fact, practitioners might be inclined to favor reliability over statistical performance [1], although it is always a trade-off that depends on the precise use case.`
>
> > There is a broad literature in statistics considering model (over-)confidence [A - C] which I think should be mentioned. In particular, power posteriors are quite popular to adjust the posterior in order to account for discrepancies between a model and real data. Since NRE does not estimate the likelihood directly, it might be necessary to use the presented approach instead of power posteriors, but a discussion should be added.
>
> Thank you for these references, we have updated the manuscript. It should be noted however that our work is not immediately concerned with the problem of model (simulator) misspecification. However, we acknowledge the importance of the problem.
>
> > “Recently, Dellaporta et al. [26] further improved GBI by combining [...]” The cited work [26] has advantages and disadvantaged but it's not an "improved" version of GBI, it's a variant. In particular, it best applies to IID data. When using time-series data as many of the models in this paper [D] would be a more fitting GBI procedure.
>
> Thanks for the clarification, we replaced `further improved` by `extended`.
>
> > “M/G/1, originally introduced by Papamakarios et al. [4], [...]” Papamakarios et al. however reference [E] for the M/G/1 model. This should be amended.
>
> Thank you for spotting this, the reference was indeed missing and has been updated.
>
> > Could you please write down the definition of a classifier? I assume you mean something like $d:X \rightarrow [0,1]$
>
> The classifier actually takes both $x$ and $\theta$ as input. The label is $1$ if the pair $(\theta, x)$ pair is sampled from the joint distribution and $0$ if it is sampled from the product of the marginals. This is now defined mathematically in the latest revision.

---

> > ### Author Response · Authors · 2022-08-01
> > **Rebuttal 2**
> >
> > > The discussion around equation (6) is confusing. You say that $\hat{p}(\theta|x)<p(\theta|x)$ whenever the $\hat{d}(\theta,x)<d(\theta,x)$, but clearly $\hat{p}(\theta|x)<p(\theta|x)$ can't hold for all $\theta$ since we are considering densities. Hence, the crucial aspects of whether the density is more conservative is for which $\theta$ one has $\hat{p}(\theta|x)<p(\theta|x)$ and that seems to be missing.
> >
> > Thank you for pointing this out. We realize our discussion should have been more thought through. We now propose the following and more complete explanation:
> >
> > Thm 1 states that ${\mathbb{E}}\_{p(x,\theta)} [d / \hat{d}] \geq 1$. This can be rewritten as ${\mathbb{E}}\_{p(x)}  \left[ {E}\_{p(\theta|x)} [d / \hat{d}] \right] \geq 1$. Ideally, if we had ${\mathbb{E}}\_{p(\theta|x)} [d / \hat{d}] \geq 1$ for all $x$, then we would have $d / \hat{d} \geq 1$ in regions of high posterior density, which would result in $\hat{d} / (1-\hat{d})  \leq d / (1-d) \Leftrightarrow \hat{r}(x \vert \theta) < r(x \vert \theta)$, hence $\hat{p}(\theta \vert x) < p(\theta \vert x)$.
> >
> > Similarly, Thm 2 states that ${\mathbb{E}}\_{p(x)p(\theta)} [(1-d) / (1-\hat{d})] \geq 1$. This can be rewritten as ${\mathbb{E}}\_{p(x)} \left[ {\mathbb{E}}\_{p(\theta)} [(1-d) / (1-\hat{d})] \right] \geq 1$. Ideally, if we had ${\mathbb{E}}\_{p(\theta)} [(1-d) / (1-\hat{d})] \geq 1$ for all $x$, then we would have $(1-d)/(1-\hat{d}) \geq 1$ in regions of high prior density, which would result in $\hat{p}(\theta \vert x) > p(\theta \vert x)$.
> >
> > Therefore enforcing the balancing condition will try to impose these two antagonistic objectives (lowering $\hat{p}(\theta \vert x)$ such that $\hat{p}(\theta \vert x) < p(\theta \vert x)$ and increasing $\hat{p}(\theta \vert x)$ such that $\hat{p}(\theta \vert x) > p(\theta \vert x)$) at the same time. Which constraint dominates will depend on whether $p(\theta|x) > p(\theta)$. If $p(\theta|x) > p(\theta)$, then the effect of Thm 1 dominates, which results in $\hat{p}(\theta \vert x) < p(\theta \vert x)$. If $p(\theta) > p(\theta|x)$ then the effect of Thm 2 dominates, which results in $\hat{p}(\theta \vert x) > p(\theta \vert x)$.
> >
> > We have now updated L124:138 to include this discussion.
> >
> > > Generally the model details are a bit short. For example, you reference Lotka and Volterra indicating that the underlying model is a deterministic ODE following the Lotka-Volterra equations. However, commonly people in SBI work with the stochastic version, describing a Markov jump process. I would be nice to have more information here.
> >
> > We use the stochastic version that is described by a Markov jump process; it is the same one as defined in Papamakarios et al. [4]. The reference was indeed missing and has now been added. Thank you for spotting this!
> >
> > > I am not so sure about the expected coverage as a special case of SBC. The internal consistency of the joint distribution that you're relying on is something that is remarked even in the SBC paper to be known before. The devil is in the detail and from my point of view SBC is about an actionable algorithm, which is more than just the expectation that you allude to in Appendix A. In particular, the actual computations (checking whether rank statistics are uniformly distributed vs comparing coverage) are quite different in both cases.
> >
> > Indeed, the way we compute coverage is quite different from the computation of rank statistics. However, the expected coverage at credibility level $1 - \alpha$
> >
> > $$ {\mathbb{E}}\_{p(\theta^\*, x^\*)} \left[ \mathbb{I} [ \theta^\* \in {\Theta}\_{\hat{p}(\theta | x^\*)}(1 - \alpha)]\right] $$
> >
> > is the probability of rank statistics
> >
> > $$ \hat{r}(\theta^*) = \mathbb{E}_{\hat{p}(\theta | x^*)} \big[  \mathbb{I} [ \hat{p}(\theta | x^\*) \leq \hat{p}(\theta^\* | x^\*)  ]  \big] $$
> >
> > to be above $\alpha$. Therefore, $1$ minus the expected coverage is the cumulative distribution function $P(\hat{r}(\theta^*) \leq \alpha)$ of the rank statistics over the joint distribution $p(\theta^*, x^*)$. Hence, we can easily recover the empirical CDF from our expected coverage plots and, thereby, the histograms (empirical PDF) of SBC. Given that the histograms of SBC can be recovered from the empirical coverage, it benefits from the interpretations linked to the SBC diagnostic, making it an actionable algorithm as well.
> >
> > That being said, we would like to insist on the fact that this link between SBC and expected coverage has been added to strengthen the use of expected coverage as a diagnostic. Never do we claim that this is a novel contribution.
> >
> > If the revised version successfully addressed all your concerns, we would like to kindly ask you to reconsider your score in light of the modifications made.

---

> > ### Comment · Reviewer_X2bu · 2022-08-03
> > **Still missing detail**
> >
> > Thank you very much for the changes. While some of the technical discussion has been improved, I have the impression that other bits have been unfairly dismissed as relatively minor.
> >
> > > Thank you for these references, we have updated the manuscript. It should be noted however that our work is not immediately concerned with the problem of model (simulator) misspecification. However, we acknowledge the importance of the problem.
> >
> > It appears that the suggested discussion is missing and I assume this is because the authors think those references are not immediately relevant, as this paper is not considering model (simulator) misspecification. However, what is considered here is that the estimate of the likelihood-to-evidence ratio is misspecified wrt to the posterior of the simulator, i.e. it is not a perfect estimate / approximation. Hence, I still believe that the relevant literature on power posteriors and coarsening [F] is highly relevant and should be discussed. After all, the paper is about finding more conservative posteriors, and there exists a literature on finding more conservative posteriors. Please do not shy away from investigating the references in those papers as well.
> >
> > > Thanks for the clarification, we replaced further improved by extended.
> >
> > I am sorry to be nitpicking, but I do not understand in what sense the method is an "extension" of GBI. Instead, it is an application of the related nonparametric learning framework [G]. (This is not a criticism of the cited work, rather it's pointing out the classification that the authors of the cited work are using. I do think it is important not to confuse potential readers by adopting a different language.)
> >
> > > Generalized Bayesian inference (GBI) extends Bayesian inference by replacing the likelihood term by an arbitrary loss function [23].
> >
> > [23] is the wrong reference for GBI, as the paper says, it is about finding the scaling parameter for a power posterior. This is relevant for this paper, but not the paper that fits this sentence. The paper that does discuss GBI as a coherent way for obtaining belief updates using general losses is [H].
> >
> > > If the revised version successfully addressed all your concerns, we would like to kindly ask you to reconsider your score in light of the modifications made.
> >
> > My initial tendency to accept the paper is based (in addition to the generally interesting idea) my belief that the weaknesses can be easily remedied by adjusting the discussed literature during the rebuttal period. I think it is essential, although remain open to be convinced otherwise, if there is something I have misunderstood.
> >
> > [F] Miller, J. W., & Dunson, D. B. (2018). Robust Bayesian inference via coarsening. Journal of the American Statistical Association.
> >
> > [G] Lyddon, S., Walker, S., & Holmes, C. C. (2018). Nonparametric learning from Bayesian models with randomized objective functions. Advances in neural information processing systems, 31.
> >
> > [23] Holmes, C. C., & Walker, S. G. (2017). Assigning a value to a power likelihood in a general Bayesian model. Biometrika, 104(2), 497-503.
> >
> > [H] Bissiri, P. G., Holmes, C. C., & Walker, S. G. (2016). A general framework for updating belief distributions. Journal of the Royal Statistical Society: Series B (Statistical Methodology), 78(5), 1103-1130.

---

> > > ### Author Response · Authors · 2022-08-05
> > > **Thanks for the additional references**
> > >
> > > Thanks for the additional references that helped us get a broader view and understand the relevance of this literature. We will update the paragraph related to model misspecification as follows to both include a discussion on power likelihood/posteriors and clarify the contributions of [30]. Please let us know if you find any mistakes in this paragraph as we are not familiar with this literature.
> > >
> > > > In this work, we make the assumption that the simulator is well-specified, in the sense that it accurately models the real data generation process. However, this assumption is often violated. To overcome this issue, Generalized Bayesian inference (GBI) extends Bayesian inference by replacing the likelihood term with an arbitrary loss function [23]. Those loss functions can be designed to mitigate specific types of misspecifications and enable robust inference, even with intractable likelihoods [24– 26]. Power likelihood losses have also been shown to increase robustness to model misspecification [27]. It consists in raising the likelihood to a power to control the impact it has over the prior. The lower the power of likelihood, the lower the importance given to the data and the higher the uncertainty of the posterior. It can either be set based on practitioner knowledge or derived from observed data [28]. Following the same objective, Miller and Dunson [29] introduce coarsened posteriors that condition on a neighborhood of the empirical data distribution rather than on the data itself. This neighborhood is derived from a distance function that, when set to the relative entropy, allows the approximation of coarsened posteriors by a power posterior. Recently, Dellaporta et al. [30] applied Bayesian non-parametric learning to GBI, making inference with misspecified simulator models both robust and computationally efficient.
> > >
> > > [23] Pier Giovanni Bissiri, Chris C Holmes, and Stephen G Walker. A general framework for
> > > updating belief distributions. Journal of the Royal Statistical Society: Series B (Statistical
> > > Methodology), 78(5):1103–1130, 2016
> > >
> > > [24] Sebastian M Schmon, Patrick W Cannon, and Jeremias Knoblauch. Generalized posteriors in
> > > approximate bayesian computation. arXiv preprint arXiv:2011.08644, 2020.
> > >
> > > [25] Takuo Matsubara, Jeremias Knoblauch, François-Xavier Briol, Chris Oates, et al. Robust
> > > generalised bayesian inference for intractable likelihoods. arXiv preprint arXiv:2104.07359,
> > > 2021.
> > >
> > > [26] Lorenzo Pacchiardi and Ritabrata Dutta. Score matched neural exponential families for
> > > likelihood-free inference. Journal of Machine Learning Research, 23(38):1–71, 2022.
> > >
> > > [27] Peter Grünwald and Thijs Van Ommen. Inconsistency of bayesian inference for misspecified
> > > linear models, and a proposal for repairing it. Bayesian Analysis, 12(4):1069–1103, 2017.
> > >
> > > [28] Chris C Holmes and Stephen G Walker. Assigning a value to a power likelihood in a general
> > > bayesian model. Biometrika, 104(2):497–503, 2017.
> > >
> > > [29] Jeffrey W Miller and David B Dunson. Robust bayesian inference via coarsening. Journal of
> > > the American Statistical Association, 2018.
> > >
> > > [30] Charita Dellaporta, Jeremias Knoblauch, Theodoros Damoulas, and François-Xavier Briol.
> > > Robust bayesian inference for simulator-based models via the mmd posterior bootstrap. In
> > > International Conference on Artificial Intelligence and Statistics, pages 943–970. PMLR, 2022.

---

> > > > ### Comment · Reviewer_X2bu · 2022-08-07
> > > > **Better but still not great**
> > > >
> > > > Thank you very much, I appreciate the changed. However, there are still many odd bits:
> > > >
> > > > > In this work, we make the assumption that the simulator is well-specified [...]
> > > >
> > > > I understand that you're in the well-specified setting, but the point I am trying to make is that the likelihood-to-evidence ratio is misspecified wrt to the simulator. In short, model misspecification occurs when $x \sim p^*$, but we evaluate it in $p(x | \theta) \neq p^*(x)$. In your case, $x \sim p(x | \theta)$, but you evaluate $\hat{d}(\theta, x)/(1- \hat{d}(\theta, x)) \neq p(x \mid \theta)/p(x)$. Hence, the approaches suggested are legitimate alternatives to obtain more conservative posteriors.
> > > > Currently, it reads as all this work has relatively little relevance to this paper.
> > > >
> > > > > Recently, Dellaporta et al. [30] applied Bayesian non-parametric learning to GBI,
> > > >
> > > > Replace GBI with LFI?

---

> > > > > ### Author Response · Authors · 2022-08-08
> > > > > **Our view on this matter**
> > > > >
> > > > > Thanks for those precisions.
> > > > >
> > > > > We understand your point and indeed model misspecification and computational unfaithfulness lead to the similar consequences and are hence connected. However, we still believe that those issues should still be viewed as separate and that the distinction between the two should be made as the causes underpinning those issues are different. The tools used for addressing them are hence also different.
> > > > >
> > > > > > Replace GBI with LFI?
> > > > >
> > > > > We have now replaced it with SBI to be consistent with the rest of the paper.

---

### Official Review · Reviewer_Vwgm · 2022-07-08

**Rating:** 6
**Confidence:** 4
**Soundness:** 3 good
**Presentation:** 3 good
**Contribution:** 3 good

**Summary:**

Simulation-based inference enables Bayesian parameter inference for simulation-based models for which the likelihood cannot be evaluated efficiently, by using simulations from the model as training data. Over the last years there has been developed a suite of neural network-based SBI methods that estimate the posterior distribution directly (neural posterior estimation, NPE), or target the likelihood (neural likelihood estimation, NLE) or the likelihood ratio (neural ratio estimation, NRE) to then obtain posterior samples via MCMC. By design, SBI is usually applied in scenarios where the ground-truth posterior is not known, and with limited training data, e.g., because model simulations are expensive. Thus, the approximated posterior (samples) may be inaccurate for a given problem. Preliminary results in the field indicate that posteriors tend to be overconfident, e.g., they tend to underestimate uncertainties.
This submission introduces a variant of NRE, called Balanced NRE (BNRE), with the goal to make posterior estimates more conservative in the low simulation budget regime. Like NRE, BNRE trains a classifier in order to approximate the likelihood-to-evidence ratio, to then perform MCMC. The authors introduce the concept of balanced classifier and introduce a modified loss function to train BNRE such it is a balanced classifier. Furthermore, they show that the optimal Bayesian classifier is balanced, i.e., that in the limit of unlimited training data BNRE will eventually converge to be NRE and obtain the exact posterior. They perform experiments with tractable benchmarking problems to show that BNRE tends to be more conservative than NRE in the low simulation budgets, i.e., show larger posterior variance. They conclude that BNRE provides a new SBI method suitable to obtain conservative posterior estimates.

**Questions:**

As outlined in detail above, I think this study would benefit from additional tasks and from additional metrics to study the effect of BNRE in more detail. For the benchmarking tasks I suggest to use one that is fully tractable so that it can be studied in detail, and one with real-world character that is challenging, e.g., high-dimensional, so that one can see the practical relevance of BNRE. For metrics I suggest to additionally use SBC, local coverage tests and analytical over and underdispersion and bias (for the tractable tasks).



**Limitations:**

The authors transparently addressed limitations of their approach, given the scope of the results presented in the paper (see comments above).

Update after rebuttal: The authors addressed all of my questions and the revised version of the submission now adequately discusses limitations of the approach to the low-dimensional parameter regime. I updated my score accordingly.

**Strengths And Weaknesses:**

**Originality:** The reliability of posterior estimates obtained with SBI is a timely and very important topic which I am happy to see addressed here. The paper picks one of three main SBI approaches, NRE, and suggests a variant with the goal to make it more conservative in the low simulation budget regimes. The introduction of balanced classifiers is novel and useful for the context of NRE and, as mentioned in the conclusion, it may indeed be applicable to NPE as well.

**Quality**: The introduction of balanced classifiers and BNRE as a variant of NRE with an additional term in the loss function is technically sound and a novelty. However, the experimental evaluation of BNRE is limited and less convincing for several reasons:
- Choice of benchmarks: The set of benchmark tasks is limited, it contains rather low-dimensional tasks most of which (all? depending on the variants used, exact definitions of the tasks missing) are tractable via MCMC. It would be more instructive to show one or two tractable examples with reference posteriors (obtained via MCMC or analytically) to study the properties of BNRE in depth, and to then show one or two additional “real-world” examples to demonstrate the use of BNRE in practice and to show how it compares to NRE in practice. I see the current results merely as a first study showing preliminary results.
- Choice of metrics: The results of the paper are mainly based on one single metric measuring the “confidence” of the posterior—the expected posterior coverage. This metric was introduced in a recent preprint by Hermans et al [1] which, again, addresses the very important topic of SBI reliability, but makes relatively strong claims about the general reliability of SBI methods based solely on the expected coverage metric, and was not successfully peer-reviewed by the community yet. The evaluation of BNRE would become more informative if additional metrics were considered. One common metric for investigating the posterior coverage and potential biases is simulation-based calibration (SBC). The authors showed in the appendix that in theory expected coverage is a special case of SBC, however, it would be useful to see the practical similarities in the results. Importantly, SBC is able to detect not only miscalibrated uncertainties like over- or underdispersion, but also positive and negative biases of the posterior estimates. Thus, it is additionally suited for evaluating BNRE given that BNRE tends to introduce a bias in the low simulation budget regimes. In addition to SBC which provides a global marginal coverage test across all x, there is a new method for performing a local coverage test for specific observation (Zhao et al. [https://proceedings.mlr.press/v161/zhao21b.html](https://proceedings.mlr.press/v161/zhao21b.html)), which would be useful to add as a metric as well, e.g., in the “real-world” example.
- For the tractable tasks with known posteriors it would additionally be useful to calculate actual biases, and dispersion with respect to the reference posteriors.
- For the real-world examples additional checks, e.g., prior and posterior predictive checks would be instructive to show the practical effect of conservative posteriors and difference of BNRE to NRE.
- Finally, a comparison to other established SBI methods like NPE and NLE would be illustrative. They are readily available in open-source software packages so that adding them to the benchmark would not result in large algorithmic or implementation overhead.

Overall I think, given the importance of this topic, it is essential to evaluate a new method on the available and established metrics (especially when additionally introducing a new metric).

**Clarity**: The paper is very well and clearly written, it was straight forward to follow the line of arguments and the presented results. Regarding the appearance of the figures I have the following comments:
- A visual explanation of the approach as Figure 0 would be nice to have
- For most of the subplots in Figure 1 it is hard to see any differences between the methods, a different visualization, e.g., on the log-scale would be more illustrative
- For Figure 2 it would be good to show standard error of the mean as error bars, given that 5 repetitions were performed
- For Figure 3 it would better to show standard error of the mean instead of the standard deviation (minor)

**Significance**: As mentioned above, I find the general topic and the approach of BNRE important for SBI research. However, the experiments and results presented here appear to me as preliminary with no direct insights or consequences for other researchers or practitioners (yet). In theory, the BNRE approach makes sense, however, in practice (as indicated by the results presented here) I do not see an advantage over NRE. The results **do** show that on the benchmark tasks presented here, BNRE leads to broader posteriors than NRE. However, the additional results in Figure 3 and 4 also indicate that it comes with a larger bias (which makes sense due to the additional term in the loss). While the bias reduces with increasing simulation budget, but even for the same budget, it seems that BNRE performs worse than NRE. Furthermore, BNRE comes with the additional hyperparameter choice for $\lambda$. The authors make a suggestion for a default value of lambda based on extensive parameter search on the benchmark tasks, however, it is not clear how this choice would extrapolate to real-world scenarios, e.g., for models with more parameters. Thus, from the practitioners point of view, it would not makes sense to use BNRE as of yet.

---

> ### Author Response · Authors · 2022-08-01
> **Rebuttal**
>
> We thank you for taking the time to review our work. We answer your main concerns below:
>
> > Like NRE, BNRE trains a classifier in order to approximate the likelihood-to-evidence ratio, to then perform MCMC.
>
> Although NRE can be used in combination with MCMC, we use here a variant of NRE that avoids the use of MCMC. We directly evaluate the approximate posterior density as $p(\vartheta)\hat{r}(\vartheta\vert x)$ over a discrete grid of parameter values.
>
> > They perform experiments with tractable benchmarking problems. [...] most of which (all? […]) are tractable [...]
>
> This is untrue, all our benchmarks have an intractable likelihood:
>
> * SLCP: As stated in the text, the likelihood is intractable due to the marginalization to infer the posterior density over 2 parameters defining the mean.
>
> * Weinberg: Simulates an experiment that measures the scattering angle of electron positron collisions. Internally, this involves a stochastic rejection loop of particle collisions based on the (random) collision cross section and beam energy.
>
> * Spatial SIR: The observable result from a sequence of random infections of individuals. This sequence of random actions makes the likelihood intractable.
>
> * M/G/1: The observable results from a sequence of random times between the arrival of customers and random times it takes to serve a customer. This sequence of random actions makes the likelihood intractable.
>
> * Lotka-Volterra: The likelihood is intractable because the population dynamics are modeled as a time series that is described by a Markov Jump Process.
>
> * Gravitational waves: The likelihood is intractable due to the marginalization over all nuisance parameters to obtain the posterior over the two masses.
>
> > The set of benchmark tasks is limited, it contains rather low-dimensional tasks.
>
> We agree that our benchmark tasks only contain low-dimensional tasks. The main reason is the coverage diagnostic becomes intractable for high-dimensional tasks and hence the benefit of BNRE cannot be empirically checked on such tasks. That being said, high-dimensional posterior inference remains a challenging issue for all SBI algorithms. We do not claim to solve this issue in our paper.
>
> > It would be more instructive to show one or two tractable examples with reference posteriors (obtained via MCMC or analytically) to study the properties of BNRE in depth, and to then show one or two additional “real-world” examples to demonstrate the use of BNRE in practice and to show how it compares to NRE in practice
>
> Thanks for the suggestion, we agree with this view. Actually, we adopt the same template but modify the order of appearance. We first start by showcasing that the method works on real-world examples in Section 4.1 and subsequently study the properties in-depth in Section 4.2.
>
> For the reference posteriors, they cannot be obtained via MCMC or analytically due to the intractable likelihood of the benchmark. However, we suggest discussing results of Section 4.2 with respect to a reference posterior obtained via NRE with a high simulation budget (e.g., the run using $2^{17}=131\text{k}$ simulations). Figure 4 is already ready as is and we will add the reference posterior on the two leftmost plots of Figure 5 by the end of the discussion period.
>
> > I see the current results merely as a first study showing preliminary results.
>
> We respectfully disagree with this statement. As stated by other reviewers, our claims are backed by theoretical arguments supported with empirical evidence. We provide evidence that BNRE leads to more conservative posteriors on a wide range of benchmarks that are representative of real world scientific applications. Moreover, we do provide insights about the behavior of the method by computing the bias and variance of the approximate posteriors and through a study of the effect of the $\lambda$ parameter. The full set of results is provided in the supplementary materials.
>
> > Choice of metrics: The results of the paper are mainly based on one single metric measuring the “confidence” of the posterior—the expected posterior coverage.
>
> We evaluate our results across 5 metrics: expected coverage, coverage AUC, expected posterior density, as well as in terms of approximation bias and variance on all benchmarks. The full set of results is provided in the supplementary materials. We frame our discussion and impact of our method based on the evaluation of all those metrics.
>
> > there is a new method for performing a local coverage test for specific observation (Zhao et al. https://proceedings.mlr.press/v161/zhao21b.html), which would be useful to add as a metric as well, e.g., in the “real-world” example.
>
> Our claims are all about the expected coverage and not the local coverage.

---

> > ### Author Response · Authors · 2022-08-01
> > **Rebuttal 2**
> >
> > > Importantly, SBC is able to detect not only miscalibrated uncertainties like over- or underdispersion, but also positive and negative biases of the posterior estimates.
> >
> > Indeed, but we already discuss the approximation bias and variance in Section 4.2 and in Appendix G. Therefore, we believe that the added value of SBC would be minor.
> >
> > In addition, we have shown in Appendix A that the expected coverage closely relates to SBC. The expected coverage at credibility level $1 - \alpha$
> >
> > $$ {\mathbb{E}}\_{p(\theta^{\*}, x^{\*})}  \left[ \mathbb{I}  [\theta^{\*} \in {\Theta}\_{\hat{p}(\theta | x^*)} (1 - \alpha) ]  \right]  $$
> >
> > is the probability of rank statistics
> >
> > $$ \hat{r}(\theta^*) = \mathbb{E}_{\hat{p}(\theta | x^*)} \big[  \mathbb{I} [ \hat{p}(\theta | x^*) \leq \hat{p}(\theta^* | x^*)  ]  \big] $$
> >
> > to be above $\alpha$. Therefore, $1$ minus the expected coverage is the cumulative distribution function $P(\hat{r}(\theta^*) \leq \alpha)$ of the rank statistics over the joint distribution $p(\theta^*, x^*)$. Hence, we can easily recover the empirical CDF from our coverage plots and, thereby, the histograms (empirical PDF) of SBC.
> >
> > This implies that the expected coverage curves can also be used to probe for negative and positive biases, further demonstrating the equivalence between both diagnostics. That being said, we would like to insist on the fact that this link between SBC and expected coverage has been added to strengthen the use of expected coverage as a diagnostic.
> >
> > > For the tractable tasks with known posteriors it would additionally be useful to calculate actual biases, and dispersion with respect to the reference posteriors.
> >
> > We agree that it could be a nice addition. We did not compute those against reference posteriors because they are unavailable in settings where the likelihood is intractable (which is the case for all our benchmarks). However, if we consider NRE with the highest simulation budget as a reference posterior, Figure 4 gives us insights by comparing the red curve (BNRE) with the last value for the blue curve (the chosen reference posterior). Moreover, we do compute and report approximation biases and variances in Appendix G. To highlight this we added, close to Figure 4, the following sentence
> >
> > *The bias gets close to $0$ for high simulation budgets, showing that the bias induced by BNRE vanishes as the simulation-budget increases.*
> >
> > > For the real-world examples additional checks, e.g., prior and posterior predictive checks would be instructive to show the practical effect of conservative posteriors and difference of BNRE to NRE.
> >
> > We agree that these might be instructive to show the practical effect and leave it as future work. However, we would like to point out that in the current version, the metrics considered are sufficient to demonstrate our claims regarding the behavior of BNRE in contrast to NRE.
> >
> > > Finally, a comparison to other established SBI methods like NPE and NLE would be illustrative. They are readily available in open-source software packages so that adding them to the benchmark would not result in large algorithmic or implementation overhead.
> >
> > [1] already demonstrates that NPE, NRE and SNL can all be overconfident. Meaning, showing that BNRE leads to conservative approximate posterior for all the benchmarks with respect to NRE is therefore a sufficient comparison. Comparing BNRE to NPE and NLE in terms of bias, variance, and log densities is, in our opinion, irrelevant for the claims made in the paper because the balancing condition is (currently) designed for SBI techniques that employ classifiers as neural surrogates for the statistical quantity of interest.
> >
> > > For Figure 2 it would be good to show standard error of the mean as error bars, given that 5 repetitions were performed
> >
> > Standard deviations (albeit not standard errors) are shown in Appendix F for the coverage AUCs. We made separate plots to avoid cluttering Figure 2.
> >
> > > For Figure 3 it would better to show standard error of the mean instead of the standard deviation (minor)
> >
> > We always consider single models and not ensembles. We do not understand the motivation for considering the standard error of the mean over 5 models instead of the standard error of the statistic computed on one model, could you elaborate further on this particular point?

---

> > > ### Author Response · Authors · 2022-08-01
> > > **Rebuttal 3**
> > >
> > > >  However, the experiments and results presented here appear to me as preliminary with no direct insights or consequences for other researchers or practitioners (yet). In theory, the BNRE approach makes sense, however, in practice (as indicated by the results presented here) I do not see an advantage over NRE. [...] it would not makes sense to use BNRE as of yet.
> > >
> > > This opinion completely ignores the theoretical arguments and the empirical evidence presented in the paper. BNRE is an algorithm that enables conservative approximations for every simulation budget essentially for free in terms of computational cost and tuning, while retaining the same global minimum as NRE!
> > >
> > > In addition, although not the main focus of our paper, the core theoretical arguments that are the foundation to BNRE have an impact on the broader AI community, because they relate to any binary classifier and can therefore be applied to any high-risk classification problem while again, sharing the same global minimum as the Bayes optimal classifier.
> > >
> > > —
> > >
> > >
> > > We hope the responses put forward here are convincing and answer your main concern regarding the validity of our theoretical arguments and empirical results. For this reason we kindly ask you to reconsider your score.

---

> > > > ### Comment · Reviewer_Vwgm · 2022-08-07
> > > > **Response to rebuttal**
> > > >
> > > > I thank the authors for taking the time to answer my remarks and questions in detail! I consider most of the minor points addressed, however, one major concern remains.
> > > > Regarding the benchmarks: thank you for the clarification, I indeed missed the point that all benchmarks are intractable, and by design so low-dimensional that evaluation of the posterior on a grid is feasible. However, this does not alleviate my concerns: I agree that the theoretical argument for BNRE made in this paper is valuable, but I find the empirical evidence that it works in practice too thin.
> > > > Again, the results show that, as intended, BNRE results in broader / more conservative posteriors in the low simulation budget-regime. However, the results in section 4 showed as well that BNRE comes with a bias in the posterior estimate. In practice, SBI problems will likely have higher dimensionality than the benchmarks presented here, thus, I find it highly problematic to present BNRE as a valid alternative to other SBI methods, without knowing how it behaves in such a scenario. I therefore think that it would be essential to evaluate BNRE with additional experiments, e.g., show on a fully tractable example how NRE and BNRE compare in terms of bias, variance and coverage as one increases the simulation budget and the dimensionality of the inference problem.

---

> > > > > ### Author Response · Authors · 2022-08-08
> > > > > **Low dimensional parameter space is interesting**
> > > > >
> > > > > Thanks for acknowledging that all the points raised but one have successfully been addressed. Regarding this last point, we disagree with the following statement:
> > > > >
> > > > > > In practice, SBI problems will likely have higher dimensionality than the benchmarks presented here, thus, I find it highly problematic to present BNRE as a valid alternative to other SBI methods, without knowing how it behaves in such a scenario.
> > > > >
> > > > > Many use cases exist for SBI in the low parameter dimensionality setting, either because the problems are low dimensional or because scientists are interested in the marginal posterior density of a small subset of parameters. All the benchmarks in this paper with the exception of SLCP are representative of actual scientific use cases. For example, the gravitational wave benchmark has led to many works applying SBI to infer the marginals over a few parameters of interest, see for example [A, B, C, D]. Other examples include studies on the dark matter aiming to infer its mass [E, F], inferring a subset of lens parameters from strong lensing [G], the Hubble constant [H], the matter density and its fluctuation in the Early Universe from weak-lensing [I]. The method presented in this manuscript is hence already useful for many real-world problems.
> > > > >
> > > > > [A] Green, S. R., Simpson, C., & Gair, J. (2020). Gravitational-wave parameter estimation with autoregressive neural network flows. Physical Review D, 102(10), 104057.
> > > > >
> > > > > [B] Dax, M., Green, S. R., Gair, J., Macke, J. H., Buonanno, A., & Schölkopf, B. (2021). Real-time gravitational wave science with neural posterior estimation. Physical review letters, 127(24), 241103.
> > > > >
> > > > > [C] Gabbard, H., Messenger, C., Heng, I. S., Tonolini, F., & Murray-Smith, R. (2022). Bayesian parameter estimation using conditional variational autoencoders for gravitational-wave astronomy. Nature Physics, 18(1), 112-117.
> > > > >
> > > > > [D] Delaunoy, A., Wehenkel, A., Hinderer, T., Nissanke, S., Weniger, C., Williamson, A. R., & Louppe, G. (2020). Lightning-fast gravitational wave parameter inference through neural amortization. arXiv preprint arXiv:2010.12931.
> > > > >
> > > > > [E] Montel, N. A., Coogan, A., Correa, C., Karchev, K., & Weniger, C. (2022). Estimating the warm dark matter mass from strong lensing images with truncated marginal neural ratio estimation. arXiv preprint arXiv:2205.09126.
> > > > >
> > > > > [F] Hermans, J., Banik, N., Weniger, C., Bertone, G., & Louppe, G. (2021). Towards constraining warm dark matter with stellar streams through neural simulation-based inference. Monthly Notices of the Royal Astronomical Society, 507(2), 1999-2011.
> > > > >
> > > > > [G]Chianese, M., Coogan, A., Hofma, P., Otten, S., & Weniger, C. (2020). Differentiable strong lensing: uniting gravity and neural nets through differentiable probabilistic programming. Monthly Notices of the Royal Astronomical Society, 496(1), 381-393.
> > > > >
> > > > > [H] Gerardi, F., Feeney, S. M., & Alsing, J. (2021). Unbiased likelihood-free inference of the Hubble constant from light standard sirens. Physical Review D, 104(8), 083531.
> > > > >
> > > > > [I] Kilbinger, M., Ishida, E. E., & Cisewski-Kehe, J. (2021). Sidestepping the inversion of the weak-lensing covariance matrix with Approximate Bayesian Computation. arXiv preprint arXiv:2112.03148.

---

> > > > > > ### Comment · Reviewer_Vwgm · 2022-08-09
> > > > > > **Response**
> > > > > >
> > > > > > Thank you for the pointers to the low-dimensional SBI problems---I agree, there are many interesting SBI problems in this regime. Given the overall rebuttal of your submission and your revised manuscript (*) I now see how BNRE can be useful in this regime of low-dimensional SBI problems, in order to obtain more conservative posteriors in the low simulation-budget regime. I am willing to adapt my score accordingly.
> > > > > > However, I would ask you to add this limitation of BNRE--that it was developed for and tested in the low-dimensional SBI regime, i.e., for <= 3D benchmarks for which (B)NRE-posteriors can be obtained by evaluating a grid--should be added to this discussion too.
> > > > > >
> > > > > > (*) the revised version breaks the 9-page limit

---

> > > > > > > ### Author Response · Authors · 2022-08-09
> > > > > > > **Thanks, the paper has been updated**
> > > > > > >
> > > > > > > We have now updated the limitations in Section 6 to reflect this:
> > > > > > >
> > > > > > > > Third, the benefits of BNRE remain to be assessed in high-dimensional parameter spaces. In particular, the posterior density must be evaluated on a discretized grid over the parameter space to compute credibility regions, which currently prohibits the accurate computation of expected coverage in the high-dimensional setting.
> > > > > > >
> > > > > > > Finally, we also want to thank you for your positive feedback and the constructive discussion that helped us improve the paper.
> > > > > > >
> > > > > > > The latest revision of the manuscript now integrates changes requested and discussed with all four reviewers, all within the 9-page limit.

---

### Official Review · Reviewer_NmkK · 2022-07-11

**Rating:** 6
**Confidence:** 4
**Soundness:** 3 good
**Presentation:** 3 good
**Contribution:** 3 good

**Summary:**

This study builds on a previous state-of-the-art method for simulation-based inference (i.e., NRE), and proposes an improvement that aims at avoiding the usual overconfidence of NRE posterior estimates (an issue that might be common to other simulation-based inference methods). The study provides empirical results on several benchmark problems and shows that the new method (BNRE) clearly improves in reliability compared to NRE as assessed by several metrics.

**Questions:**

As pointed out above, I believe it would be beneficial to the manuscript if evidence was provided for the statement that $\lambda$ = 100 is a good choice across benchmark problems. An appendix figure would be enough.


**Limitations:**

The authors have adequately addressed the limitations of their work.

**Strengths And Weaknesses:**

### Originality

To the best of my knowledge, this is one of two studies that specifically addresses the issue of overconfidence of simulation-based inference algorithms, the other one being Hermans et al. 2021 (cited by the authors). While Hermans et al. diagnose this problem on several simulation-based inference methods and provide a partial solution to this issue using ensemble approaches, the solution provided by this study is novel.


### Quality

The paper is technically sound, and most claims are backed by empirical evidence. However, there is one claim that I would appreciate the authors to illustrate: "We found  $\lambda$ = 100 to perform well across all benchmarks, which again, is supported by Figure 5." I would recommend the authors to provide a figure (in appendix) illustrating this, even if for a subset of the benchmark problems.


### Clarity

The manuscript is clearly written, providing enough information to understand the technical contribution and empirical results, while appropriately putting the contributions in the context of previous work. A few small comments:

-page 4, line 119, "does not modify the global optimum". This is a bit imprecise and might lead to confusion, so I would suggest the authors to expand on it;

-page 4, line 122, "[...] result in increasingly conservative [...]". By "increasingly", I assume the authors meant with increasing $\lambda$. As this is not clear, I would suggest the authors to drop the word "increasingly";

-page 4, line 123, the word "Ideally" wrongly suggests that the expression (6) only follows from the previous inequality in the ideal case, which is not what the authors meant. Dropping the word "Ideally" would solve the issue;

-page 7, line 186, for clarity, the notation for nominal parameter ($\vartheta^*$) should be introduced close to the expected squared error over the approximate posterior.


### Significance

The technical development and empirical improvement of BNRE over NRE will be of interest to the ML community, in particular to the simulation-based inference community. Notably, the technical contribution could directly inspire the improvement of other methods for simulation-based inference.

---

> ### Author Response · Authors · 2022-08-01
> **Rebuttal**
>
> Thank you for your positive review and for pointing out the novelty, good validation, and significance of BNRE.
>
> > The paper is technically sound, and most claims are backed by empirical evidence. However, there is one claim that I would appreciate the authors to illustrate: "We found λ = 100 to perform well across all benchmarks, which again, is supported by Figure 5." I would recommend the authors to provide a figure (in appendix) illustrating this, even if for a subset of the benchmark problems.
>
> This sentence should not be viewed as a strong claim that the value $\lambda = 100$ performs best across all benchmarks but rather as a reasonably good default value. To obtain the best value for $\lambda$, we recommend to start with a low value for $\lambda$ and gradually increase $\lambda$ until the approximate posterior becomes conservative (e.g., when its coverage AUC becomes positive). This value is expected to fluctuate across problems. The advantage of following this procedure is that it maximizes the statistical performance of the posterior estimator while ensuring it is conservative.
>
> In the latest revision, we have now replaced the sentence
>
>  *In practice, $\lambda$ should be sufficiently large such that the approximate classifier is balanced, while maximizing the statistical performance of the posterior estimator. We found $\lambda = 100$ to perform well across all benchmarks, which again, is supported by Figure 5.*
>
> by
>
> *In practice, $\lambda$ should be sufficiently large such that the approximate classifier is balanced, while maximizing the statistical performance of the posterior estimator. Therefore, we recommend starting with a small value for $\lambda$ and to gradually increase $\lambda$ until the posterior estimator becomes conservative. We empirically found $\lambda=100$ to be a reasonably good default value leading to good performance across all considered benchmarks with various model architectures.*
>
> > page 4, line 119, "does not modify the global optimum". This is a bit imprecise and might lead to confusion, so I would suggest the authors to expand on it;
>
> We have now replaced this sentence by
>
> *Therefore, minimizing the cross-entropy loss while restricting the model hypothesis space to balanced classifiers results in the same Bayes optimal classifier of Eqn. 1.*
>
> > page 4, line 122, "[...] result in increasingly conservative [...]". By "increasingly", I assume the authors meant with increasing λ. As this is not clear, I would suggest the authors to drop the word "increasingly";
>
> We have now dropped the word “increasingly”. Thank you for your suggestion.
>
> > page 7, line 186, for clarity, the notation for nominal parameter ($\theta^*$) should be introduced close to the expected squared error over the approximate posterior.
>
> This is now addressed in the latest revision.
>
> If the revised version successfully addressed all your concerns, we would like to kindly ask you to reconsider your score in light of the modifications made.

---

### Official Review · Reviewer_oxwn · 2022-07-12

**Rating:** 6
**Confidence:** 4
**Soundness:** 4 excellent
**Presentation:** 4 excellent
**Contribution:** 3 good

**Summary:**

The authors propose a modification to the neural ratio estimation (NRE) algorithm in the form of a regularization penalty to avoid overconfident posterior distributions. The authors support their claim by introducing a family of likelihood-to-evidence ratio classifiers which are more conservative than the optimal Bayes classifier, in expectation. By posing an explicit penalty on overconfidence, the authors argue that the simulation-based inference using the NRE algorithm would be more reliable and not leading to false inference. The authors provide empirical results over several datasets, showing how the balanced neural ratio estimation (BNRE) is indeed experimentally conservative and also converges ultimately to the same posterior as the NRE method for large simulation budgets.

**Questions:**

I have no main questions for the authors, apart from any feedback on my comments above.

**Limitations:**

The one (minor) limitation I would like to flag is the various benchmarks description being relegated to Appendix C.
I believe it would be nice for the reader to appreciate the range of the benchmarks included in the paper, with some of the simulators being not necessarily trivial. Without a description, even a quick one, a non-expert reader might see these benchmarks as all simple toy-examples.

**Strengths And Weaknesses:**

The paper is well written and well presented, with legible figures and thoughtful considerations. I have checked the proofs of Theorems 1 and 2 and they look correct to me. The validation is extensive and well thought of.

The minor weakness that I can see are (a) the applicability to the NRE algorithm only (although other algorithms are mentioned in the discussion) and (b) examples of over-confident simulation-based inference algorithms leading a research direction astray.
For (a), as shown by [1], there does not seem to exist a universally "best" algorithm in SBI, so potentially extending the same logic to more algorithms would be very valuable for the community.
For (b), I appreciate the empirical results pointed out by [2]; my only point is that a practical example of a "failure case" -- or even a "if they used this overconfident SBI algorithm in this real world study it would have been a problem" type-of-argument -- would make this paper much more convincing.

Overall, I believe the paper proposes an interesting idea and it is still worth flagging in the community, although I believe that it is not as compelling as it could be.

[1] Benchmarking Simulation-Based Inference, Lueckmann et al, AISTATS 2021
[2] Averting A Crisis In Simulation-Based Inference, Hermans et al, 2021

---

> ### Author Response · Authors · 2022-08-01
> **Rebuttal**
>
> Thank you for the insightful and constructive review. We appreciate that you find our validation extensive and well thought of.
>
> > (a) There does not seem to exist a universally "best" algorithm in SBI, so potentially extending the same logic to more algorithms would be very valuable for the community.
>
> We agree that it would have been a nice addition to extend our approach to other inference algorithms. Actually, we tried to enforce the balancing condition to NPE by expressing the modeled posterior through a binary classifier, but ended up with unsatisfactory results. After trying to solve this issue for some time, we believe that the reason behind those unsatisfactory results is not trivial and decided to leave it as future work. Our current hypothesis is that a regular classifier has more flexibility to satisfy the balancing condition in contrast to a flow, since it is not constrained to be a proper density by construction.
>
> > (b) [...] my only point is that a practical example of a "failure case" -- or even a "if they used this overconfident SBI algorithm in this real world study it would have been a problem" type-of-argument -- would make this paper much more convincing.
>
> This is indeed a good suggestion to make the motivation more compelling. In the upcoming revision, we will include an example that relates to Dark Matter studies. Of particular interest to these studies is determining the “Dark Matter model” of our universe: which could be cold, warm or hot dark matter. These models describe how clumps of Dark Matter (so-called subhalos) are distributed in the universe. Cold dark matter refers to a distribution of Dark Matter that contains smaller, and more clumpy dark matter subhalos, whereas the gravitational field in hot dark matter is very smooth. In general, thermal dark matter models can be described by a single parameter, the dark matter thermal relic mass, which can be intuitively thought of as the energy the dark matter particle had in the Early Universe. A low particle energy corresponds to a warm or hot dark matter model, while a relatively high particle energy is descriptive of cold dark matter.
>
> Previously, astronomers mainly relied on rejection sampling studies (rejection ABC) to determine a posterior of the dark matter thermal relic mass (energy). However, these studies mostly relied on hand-crafted summary statistics based on the insights of astronomers. With the advent of Deep Learning and modern SBI techniques, the relation between model parameters and simulated data is automatically learned to produce approximate posteriors. While these techniques typically improve upon the obtained constraints, their explainability remains lacking. Suppose that for some reason practitioners apply an SBI algorithm without diagnosing the learned estimator. In that case, it is possible that they obtain a constraint that is smaller than it should be. Whenever an overconfident estimator produces posterior estimates that favor cold dark matter models, it could easily wipe out decades of research on the Sterile Neutrino, a potential candidate for the Warm Dark Matter particle.
>
> Of course, the above holds under the assumption where the simulator is correctly specified. Whenever the model is misspecified, which is most likely the case anyway, the problem becomes more challenging. On the other hand, combining the intuition of astronomers and a conservative posterior estimator could steer the development of their scientific model in the right direction, especially if the conservative estimator produces posterior approximations that exclude hypotheses that are consistent with observation or theory.
>
> We will include this example in the manuscript by the end of the discussion period.
>
> > The one (minor) limitation I would like to flag is the various benchmarks description being relegated to Appendix C. I believe it would be nice for the reader to appreciate the range of the benchmarks included in the paper, with some of the simulators being not necessarily trivial. Without a description, even a quick one, a non-expert reader might see these benchmarks as all simple toy-examples.
>
> Thank you for your suggestion. We have rewritten the sentence to be `We evaluate the expected coverage of posterior estimators produced by both NRE and BNRE on various problems, whose descriptions can be found in Appendix C` by `We evaluate the expected coverage of posterior estimators produced by both NRE and BNRE on various problems. Those benchmarks cover a diverse set of problems from epidemiology (Spatial SIR), astronomy (Gravitational Waves), particle physics (Weinberg) and population dynamics (Lotka Volterra). They are representative of real scientific applications of simulation-based inference. A more detailed description of the benchmarks can be found in Appendix C`.
>
> If the revised version successfully addressed all your concerns, we would like to kindly ask you to reconsider your score in light of the modifications made.

---

### Author Response · Authors · 2022-08-01
**General comment**

First and foremost we would like to thank the reviewers for the high quality of their reviews and the positive reception of our work regarding its significance, originality and experimental rigor. We appreciate the suggestions to improve the presentation of our work. All of them will be implemented and submitted by the end of the discussion period.

---

### Meta-Review · Area_Chair_AE8h · 2022-08-23

**Recommendation:** Accept
**Confidence:** Certain

**Metareview:**

The paper proposes a modification to the neural ratio estimation algorithm in the context of SBI (simulation-based inference) that tends to avoid overconfident posteriors. This is important for applications (for example in scientific discovery) where excluding plausible inferences can be more detrimental than including implausible ones.

The reviewers found the paper to be well written, technically solid, and a useful contribution to the SBI literature. Most concerns were addressed during the discussion period, with the paper strengthening its discussion of limitations as a result. In the end, the reviewers unanimously awarded the paper a score of 6 (weak accept). Therefore, I'm happy to recommend this paper for acceptance.

**Award:**

No

---

### Decision · Program_Chairs · 2022-09-14

Accept